# DIFFUSION MODELS ARE EVOLUTIONARY ALGORITHMS

**Yanbo Zhang**[1*]  **Benedikt Hartl**[1,2*]  **Hananel Hazan**[1†*]  **Michael Levin**[1,3]
[1] Allen Discovery Center at Tufts University, Medford, MA, 02155, USA
[2] Institute for Theoretical Physics, TU Wien, Austria
[3] Wyss Institute for Biologically Inspired Engineering at Harvard University,
  Boston, MA, 02115, USA

## ABSTRACT

In a convergence of machine learning and biology, we reveal that diffusion models are evolutionary algorithms. By considering evolution as a denoising process and reversed evolution as diffusion, we mathematically demonstrate that diffusion models inherently perform evolutionary algorithms, naturally encompassing selection, mutation, and reproductive isolation. Building on this equivalence, we propose the Diffusion Evolution method: an evolutionary algorithm utilizing iterative denoising – as originally introduced in the context of diffusion models – to heuristically refine solutions in parameter spaces. Unlike traditional approaches, Diffusion Evolution efficiently identifies multiple optimal solutions and outperforms prominent mainstream evolutionary algorithms. Furthermore, leveraging advanced concepts from diffusion models, namely latent space diffusion and accelerated sampling, we introduce Latent Space Diffusion Evolution, which finds solutions for evolutionary tasks in high-dimensional complex parameter space while significantly reducing computational steps. This parallel between diffusion and evolution not only bridges two different fields but also opens new avenues for mutual enhancement, raising questions about open-ended evolution and potentially utilizing non-Gaussian or discrete diffusion models in the context of Diffusion Evolution.

## 1 INTRODUCTION

At least two processes in the biosphere have been recognized as capable of generalizing and driving novelty: evolution, a slow variational process adapting organisms across generations to their environment through natural selection (Darwin, 1959; Dawkins, 2016); and learning, a faster transformational process allowing individuals to acquire knowledge and generalize from subjective experience during their lifetime (Kandel, 2013; Courville et al., 2006; Holland, 2000; Dayan & Abbott, 2001). These processes are intensively studied in distinct domains within artificial intelligence. Relatively recent work has started drawing parallels between the seemingly unrelated processes of evolution and learning (Watson & Levin, 2023; Vanchurin et al., 2022; Levin, 2022; Watson et al., 2022; Kouvaris et al., 2017; Watson & Szathmáry, 2016; Watson et al., 2016; Power et al., 2015; Hinton et al., 1987; Baldwin, 2018). We here argue that in particular diffusion models (Sohl-Dickstein et al., 2015; Song et al., 2020b; Ho et al., 2020; Song et al., 2020a), where generative models trained to sample data points through incremental stochastic denoising, can be understood through evolutionary processes, inherently performing natural selection, mutation, and reproductive isolation.

The evolutionary process is fundamental to biology, enabling species to adapt to changing environments through mechanisms like natural selection, genetic mutations, and hybridizations (Rosen, 1991; Wagner, 2015; Dawkins, 1996); this adaptive process introduces variations in organisms' genetic codes over time, leading to well-adapted and diverse individuals (Mitchell & Cheney, 2024; Levin, 2023; Gould, 2002; Dennett, 1995; Smith & Szathmary, 1997; Szathmáry, 2015). Evolutionary algorithms utilize such biologically inspired variational principles to iteratively refine sets of numerical parameters that encode potential solutions to often rugged objective functions (Vikhar,

---

*Equal contributions. [†] Author of correspondence: `Hananel@Hazan.org.il`

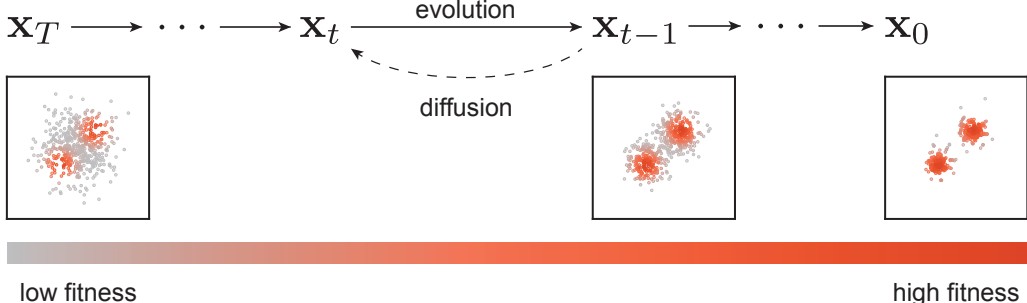

Figure 1: Evolution processes can be viewed as the inverse process of diffusion, where higher fitness populations (red points) have higher final probability density. The initially unstructured parameters are iteratively refined towards high-fitness regions in parameter space.

2016; Golberg, 1989; Grefenstette, 1993; Holland, 1992). Notably, recent breakthroughs in deep learning have led to the development of diffusion models–generative algorithms that iteratively refine data points to sample novel yet realistic data following complex target distributions: models like Stable Diffusion (Rombach et al., 2022) and Sora (Brooks et al., 2024) demonstrate remarkable realism and diversity in generating image and video. Notably, both evolutionary processes and diffusion models rely on iterative refinements that combine directed updates with undirected perturbations: in evolution, random genetic mutations introduce diversity while natural selection guides populations toward greater fitness, and in diffusion models, random noise is progressively transformed into meaningful data through learned denoising steps that steer samples toward the target distribution. This parallel raises fundamental questions: Are the mechanisms underlying evolution and diffusion models fundamentally connected? Is this similarity merely an analogy, or does it reflect a deeper mathematical duality between biological evolution and generative modeling?

To answer these questions, we first examine evolution from the perspective of generative models. By considering populations of species in the biosphere, the variational evolution process can also be viewed as a transformation of distributions: the distributions of genotypes and phenotypes. Over evolutionary time scales, mutation and selection collectively alter the shape of these distributions. Similarly, many biologically inspired evolutionary algorithms can be understood that way: they optimize an objective function by maintaining and iteratively changing a large population's distribution. This concept is central to most generative models: the transformation of distributions. Variational Autoencoders (VAEs) (Kingma, 2013), Generative Adversarial Networks (GANs) (Goodfellow et al., 2014), and diffusion models are all trained to transform simple distributions, typically standard Gaussian distributions, into complex distributions, where the samples represent meaningful images, videos, or audio, etc.

On the other hand, diffusion models can also be viewed from an evolutionary perspective. As a generative model, diffusion models transform Gaussian distributions in an iterative manner into complex, structured data points that resemble the training data distribution. During the training phase, the data points are corrupted by adding noise, and the model is trained to predict this added noise to reverse the process. In the sampling phase, starting with Gaussian-distributed data points, the model iteratively denoises to incrementally refine the data point samples. By considering noise-free samples as the desired outcome, such a directed denoising can be interpreted as directed selection, with each step introducing slight noise, akin to mutations. Together, this resembles an evolutionary process (Fields & Levin, 2020), where evolution is formulated as a combination of deterministic dynamics (e.g., natural selection) and stochastic mutations (Ao, 2005; Ping et al., 2014). If one were to revert the time direction of an evolutionary process, the evolved population of potentially highly correlated high-fitness solutions will dissolve gradually akin to the forward process in diffusion models, into the respectively chosen initial distribution, typically Gaussian noise, see Figure 1.

Driven by this intuition, we conduct a thorough investigation into the connections between diffusion models and evolutionary algorithms, discovering that these seemingly disparate concepts share the same mathematical foundations. This insight leads to a novel approach, the Diffusion Evolution

algorithm, which directly utilizes the framework of diffusion models to perform evolutionary optimization. This can be obtained by inverting the diffusion process with the Bayesian method. Our analytical study of Diffusion Evolution reveals promising parallels to biological evolution, naturally incorporating concepts such as mutation, hybridization, and even reproductive isolation.

This equivalence provides a new way of improving evolutionary algorithms and has the potential to unify developments in both fields. Mimicking biological evolution, evolutionary algorithms have shown promising results in numerical optimization, particularly for tasks that cannot be effectively trained using gradient-based methods (Wang et al., 2024; Goodfellow et al., 2014). These algorithms thus excel in exploring complex, rugged search spaces and finding globally optimal or near-optimal solutions (Hansen, 2016; Hansen & Ostermeier, 2001; Sehnke et al., 2010). While the biosphere exhibits extreme diversity in lifeforms, many evolutionary strategies, such as CMA-ES (Hansen & Ostermeier, 2001), and PEPG (Sehnke et al., 2010), struggle to find diverse solutions (Lehman & Stanley, 2011). However, our Diffusion Evolution algorithm offers a new approach. By naturally incorporating mutation, hybridization, and reproductive isolation, our algorithm can discover diverse solutions, mirroring the diversity of the biosphere, rather than converging on a single solution as is often the case with traditional methods. Since this parallel between diffusion and evolution exists naturally and is not imposed by our design, the two fields – diffusion models and evolutionary computing – can mutually benefit from each other. For example, we demonstrate that the concepts of latent diffusion (Rombach et al., 2022) and accelerated sampling (Nichol & Dhariwal, 2021) can significantly improve the performance of our Diffusion Evolution algorithm. Here, we focus on the theoretical foundations. In a complementary contribution (Hartl et al., 2024), deep-learning-based diffusion models are employed in an analogous evolutionary setting to leverage conditional sampling for explicit guidance. This approach enables controlling desired outputs and conditioning on fitness, potentially accelerating optimization without additional shaping.

In the following sections, we will first review evolutionary strategies and diffusion models, introduce the mathematical connection between diffusion and evolution, and propose the Diffusion Evolution algorithm. Then, we will quantitatively compare our algorithm to conventional evolutionary strategies, demonstrating its capability to find multiple solutions, solve complex evolutionary tasks, and incorporate developments from diffusion model literature. Lastly, we will discuss the emerging connections between the derived algorithm and evolution, along with the potential implications of this finding and the limitations of our algorithm. Codes are available on Github[1].

## 2 BACKGROUND

### 2.1 EVOLUTIONARY ALGORITHMS

The principles of evolution extend far beyond biology, offering exceptional utility in addressing complex systems across various domains. The key components of this process – imperfect replication with heredity and fitness-based selection – are sufficiently general to find applications in diverse fields. In computer and data science, for instance, evolutionary algorithms play a crucial role in optimization (Vikhar, 2016; Grefenstette, 1993; Golberg, 1989; Holland, 1992). These heuristic numerical techniques, such as CMA-ES (Hansen & Ostermeier, 2001) and PEPG (Sehnke et al., 2010), maintain and optimize a population of genotypic parameters over successive generations through operations inspired by biological evolution, such as selection of the fittest, reproduction, genetic crossover, and mutations. The goal is to gradually adapt the parameters of the entire population in such a way that individual genotypic samples, or short individuals, perform well when evaluated against an objective- or fitness function. These algorithms harness the dynamics of evolutionary biology to discover optimal or near-optimal solutions within vast, complex, and otherwise intractable parameter spaces. The evaluated numerical fitness score of an individual correlates with its probability of survival and reproduction, ensuring that individuals with advantageous traits have a greater chance of passing their genetic information to the next generation, thus driving the evolutionary process toward more optimal solutions. Such approaches are particularly valuable when heuristic solutions are needed to explore extensive combinatorial and permutation landscapes.

Some evolutionary algorithms operate with discrete, others with continuous sets of parameters. Here, we focus on the latter since discrete tasks can be seen as a subcategory of continuous tasks.

---

[1]https://github.com/Zhangyanbo/diffusion-evolution

Typically, the structure of the parameter space is a priori unknown. Thus, the initial population is often sampled from a standard normal distribution. As explained above, this initially random population is successively adapted and refined, generation by generation, to perform well on an arbitrary objective function. Thus, initially randomized parameters are successively varied by evolutionary algorithms into sets of potentially highly structured parameters that perform well on the specified task, eventually (and hopefully) solving the designated problem by optimizing the objective function. Thus, evolutionary algorithms can be understood as generative models that use heuristic information about already explored regions of the parameter space (at least from the previous generation) to sample potentially better-adapted offspring individuals for the next generation (c.f., CMA-ES (Hansen et al., 2003), etc.).

## 2.2 DIFFUSION MODELS

Diffusion models, such as denoising diffusion probabilistic models (DDPM) (Ho et al., 2020) and denoising diffusion implicit models (DDIM) (Song et al., 2020a), have shown promising generative capabilities in image, video, and even neural network parameters (Wang et al., 2024). Similar to other generative approaches such as GANs, VAEs, and flow-based models (Dinh et al., 2016; Chen et al., 2019), diffusion models transform a simple distribution, often a Gaussian, into a more complex distribution that captures the characteristics of the training data. Diffusion models achieve this, in contrast to other techniques, via iterative denoising steps, progressively transforming noisy data into less noisy (Raya & Ambrogioni, 2024), more coherent representations (Sohl-Dickstein et al., 2015).

Diffusion models have two phases: diffusion and denoising. In the diffusion phase, we are blending original data points with some extent of Gaussian noise. Specifically, let $\boldsymbol{x}_0$ be the original data point and $\boldsymbol{x}_T$ be the fully distorted data, then the process of diffusion can be represented as:

$$\boldsymbol{x}_t = \sqrt{\alpha_t}\boldsymbol{x}_0 + \sqrt{1-\alpha_t}\boldsymbol{\epsilon}, \tag{1}$$

where the amount of total noise $\boldsymbol{\epsilon} \sim \mathcal{N}(0, I)$ added to the data $\boldsymbol{x}_0$ at time step $t \in [0, T]$ is controlled by $\alpha_t$ that is monotonously decreasing from $\alpha_0 = 1$ to $\alpha_T \sim 0$. Thus, while $\boldsymbol{x}_0$ represents the original data, $\boldsymbol{x}_T$ will consist entirely of Gaussian noise. To restore such diffused data, a predictive model, typically a neural network $\boldsymbol{\epsilon_\theta}$ with parameter $\boldsymbol{\theta}$, is trained to predict the added total noise given $\boldsymbol{x}_t$ and time step $t$. Thus, diffusion models can be trained by minimizing the loss function:

$$\mathcal{L} = \mathbb{E}_{\boldsymbol{x}_0 \sim p_{\text{data}}, \boldsymbol{\epsilon} \sim \mathcal{N}(0,I)} \|\boldsymbol{\epsilon_\theta}(\sqrt{\alpha_t}\boldsymbol{x}_0 + \sqrt{1-\alpha_t}\boldsymbol{\epsilon}, t) - \boldsymbol{\epsilon}\|^2, \tag{2}$$

where $p_{\text{data}}$ is the distribution of training data. So, conventionally, diffusion models are understood as predicting the added noise during the diffusion process.

In the denoising phase, starting with a noisy pattern, the trained models are used to iteratively remove the predicted noise from current data: from $\boldsymbol{x}_T \sim \mathcal{N}(0, I)$, iteratively refine to $\boldsymbol{x}_{T-1}, \boldsymbol{x}_{T-2}, \ldots$, until restoring the data $\boldsymbol{x}_0$. In the DDIM framework, this sampling process is given by:

$$\boldsymbol{x}_{t-1} = \sqrt{\alpha_{t-1}}\left(\frac{\boldsymbol{x}_t - \sqrt{1-\alpha_t}\boldsymbol{\epsilon_\theta}(\boldsymbol{x}_t, t)}{\sqrt{\alpha_t}}\right) + \sqrt{1-\alpha_{t-1}-\sigma_t^2}\boldsymbol{\epsilon_\theta}(\boldsymbol{x}_t, t) + \sigma_t\boldsymbol{w}, \tag{3}$$

where $\sigma_t$ controls the amount of noise $\boldsymbol{w} \sim \mathcal{N}(0, I)$ added during the denoising phase. Notably, the schedule of $\alpha_t$ and $\sigma_t$ will both affect the denoising process and can be chosen based on our needs under the DDIM framework.

## 3 DIFFUSION MODELS ARE EVOLUTIONARY ALGORITHMS

Similar to the relationship between energy and probability in statistical physics, evolutionary search can be connected to generative tasks by mapping fitness to probability density: higher fitness corresponds to higher probability density. Thus, given a fitness function $f : \mathbb{R}^n \to \mathbb{R}$, we can choose a mapping $g$ to transform $f$ into a probability density function $p(\boldsymbol{x}) = g[f(\boldsymbol{x})]$. When aligning the denoising process in a diffusion model with evolution, we want $\boldsymbol{x}_0$ to follow this density function, i.e., $p(\boldsymbol{x}_0 = \boldsymbol{x}) = g[f(\boldsymbol{x})]$. This requires an alternative view of diffusion models (Song et al., 2020a): diffusion models are directly predicting the original data samples from noisy versions of those samples at each time step. Given the diffusion process $\boldsymbol{x}_t = \sqrt{\alpha_t}\boldsymbol{x}_0 + \sqrt{1-\alpha_t}\boldsymbol{\epsilon}$, we can easily express $\boldsymbol{x}_0$ in terms of the noise $\boldsymbol{\epsilon}$, and vise versa:

$$\boldsymbol{x}_0 = \frac{\boldsymbol{x}_t - \sqrt{1-\alpha_t}\boldsymbol{\epsilon}}{\sqrt{\alpha_t}}, \text{ and } \boldsymbol{\epsilon} = \frac{\boldsymbol{x}_t - \sqrt{\alpha_t}\boldsymbol{x}_0}{\sqrt{1-\alpha_t}}. \tag{4}$$

In diffusion models, the error $\epsilon$ between $\boldsymbol{x}_0$ and $\boldsymbol{x}_t$ is estimated by a neural network, i.e., $\hat{\epsilon} = \epsilon_\theta(\boldsymbol{x}_t, t)$. Thus, Equation 4 provides an estimation $\hat{\boldsymbol{x}}_0$ for $\boldsymbol{x}_0$ when replacing $\epsilon$ with $\hat{\epsilon}$. Hence, the sampling process of DDIM (Song et al., 2020a) in Equation 3 can be written as:

$$\boldsymbol{x}_{t-1} = \sqrt{\alpha_{t-1}}\hat{\boldsymbol{x}}_0 + \sqrt{1 - \alpha_{t-1} - \sigma_t^2}\hat{\epsilon} + \sigma_t \boldsymbol{w}. \tag{5}$$

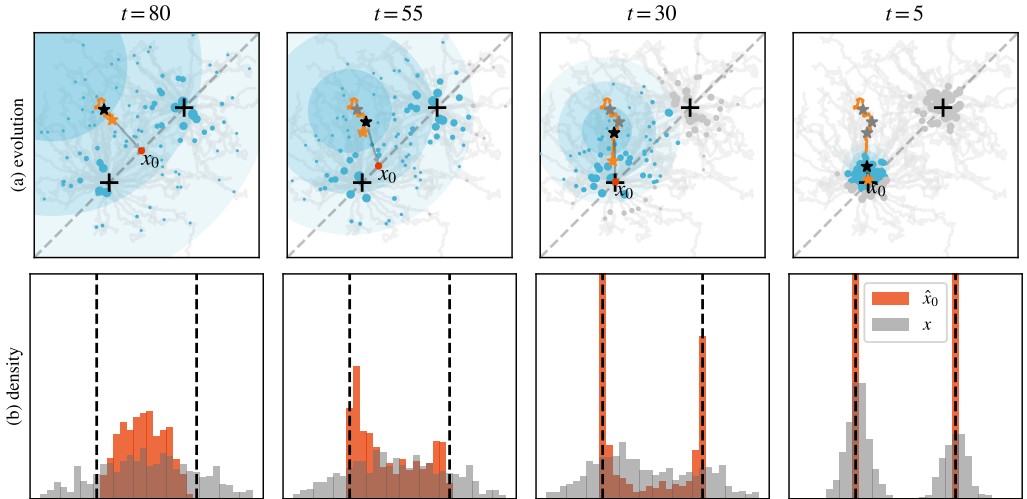

Figure 2: **(a)** Diffusion Evolution on a two-peak fitness landscape: Populations near the two black crosses have higher fitness. For each individual $\boldsymbol{x}_t$ (black star), its target $\hat{\boldsymbol{x}}_0$ (red dots) is estimated by a weighted average of its neighbors (c.f., dots within the blue disks, respectively); larger dot-size indicates higher fitness. The individual then moves a small step forward to the next generation (orange star). As evolution proceeds, the neighbor range decreases, making the process increasingly sensitive to local neighbors, thereby enabling global competition originally, while "zooming in" eventually to balance between optimization and diversity. **(b)** By mapping the population to a 1-D space (dashed lines in (a)), we track the progress of Diffusion Evolution. As evolution progresses, both the individuals (gray) and their estimated origins (red) move closer to the targets (vertical dashed lines), with the estimated origins advancing faster.

Since the denoising step in diffusion models requires an estimation of $\boldsymbol{x}_0$, we need to derive it from sample $\boldsymbol{x}_t$ and the corresponding fitness $f(\boldsymbol{x}_t)$. The estimation of $\boldsymbol{x}_0$ can be expressed as a conditional probability $p(\boldsymbol{x}_0 = \boldsymbol{x}|\boldsymbol{x}_t)$. Using Bayes' theorem and $p(\boldsymbol{x}_0 = \boldsymbol{x}) = g[f(\boldsymbol{x})]$ yields:

$$p(\boldsymbol{x}_0 = \boldsymbol{x}|\boldsymbol{x}_t) = \frac{p(\boldsymbol{x}_t|\boldsymbol{x}_0 = \boldsymbol{x})p(\boldsymbol{x}_0 = \boldsymbol{x})}{p(\boldsymbol{x}_t)} = \frac{p(\boldsymbol{x}_t|\boldsymbol{x})g[f(\boldsymbol{x})]}{p(\boldsymbol{x}_t)}. \tag{6}$$

Here, $p(\boldsymbol{x}_t|\boldsymbol{x}_0 = \boldsymbol{x})$ can be computed easily by $\mathcal{N}(\boldsymbol{x}_t; \sqrt{\alpha_t}\boldsymbol{x}, 1 - \alpha_t)$ given the design of the diffusion process, i.e., $\boldsymbol{x}_t = \sqrt{\alpha_t}\boldsymbol{x}_0 + \sqrt{1 - \alpha_t}\epsilon$. Since deep-learning-based diffusion models are trained using mean squared error loss, the $\boldsymbol{x}_0$ estimated by $\boldsymbol{x}_t$ should be the weighted average of the sample $\boldsymbol{x}$. Hence, the estimation function of $\boldsymbol{x}_0$ becomes:

$$\hat{\boldsymbol{x}}_0(\boldsymbol{x}_t, \boldsymbol{\alpha}, t) = \sum_{\boldsymbol{x} \sim p_{\text{eval}}(\boldsymbol{x})} p(\boldsymbol{x}_0 = \boldsymbol{x}|\boldsymbol{x}_t)\boldsymbol{x} = \sum_{\boldsymbol{x} \sim p_{\text{eval}}(\boldsymbol{x})} g[f(\boldsymbol{x})]\frac{\mathcal{N}(\boldsymbol{x}_t; \sqrt{\alpha_t}\boldsymbol{x}, 1 - \alpha_t)}{p(\boldsymbol{x}_t)}\boldsymbol{x}, \tag{7}$$

where $p_{\text{eval}}$ is the evaluation sample on which we compute the fitness score, here given by the current population $\boldsymbol{X}_t = (\boldsymbol{x}_t^{(1)}, \boldsymbol{x}_t^{(2)}, ..., \boldsymbol{x}_t^{(N)})$ of $N$ individuals. Equation 7 has three weight terms: The first term $g[f(\boldsymbol{x})]$ assigns larger weights to high fitness samples. For each individual sample $\boldsymbol{x}_t$, the second Gaussian term $\mathcal{N}(\boldsymbol{x}_t; \sqrt{\alpha_t}\boldsymbol{x}, 1 - \alpha_t)$ makes each individual only sensitive to local neighbors of evaluation samples. The third term $p(\boldsymbol{x}_t)$ is a normalization term. Hence, $\hat{\boldsymbol{x}}_0$ can be simplified to:

$$\hat{\boldsymbol{x}}_0(\boldsymbol{x}_t, \boldsymbol{\alpha}, t) = \frac{1}{Z}\sum_{\boldsymbol{x} \in \boldsymbol{X}_t} g[f(\boldsymbol{x})]\mathcal{N}(\boldsymbol{x}_t; \sqrt{\alpha_t}\boldsymbol{x}, 1 - \alpha_t)\boldsymbol{x}, \tag{8}$$

where $Z$ is the normalization term:

$$Z = p(\boldsymbol{x}_t) = \sum_{\boldsymbol{x} \in \boldsymbol{X}_t} g[f(\boldsymbol{x})] \mathcal{N}(\boldsymbol{x}_t; \sqrt{\alpha_t}\boldsymbol{x}, 1 - \alpha_t). \tag{9}$$

When substituting Equation 8 into Equation 4 we can express $\hat{\boldsymbol{\epsilon}}$ as:

$$\hat{\boldsymbol{\epsilon}}(\boldsymbol{x}_t, \boldsymbol{\alpha}, t) = \frac{\boldsymbol{x}_t - \sqrt{\alpha_t}\,\hat{\boldsymbol{x}}_0(\boldsymbol{x}_t, \boldsymbol{\alpha}, t)}{\sqrt{1 - \alpha_t}}, \tag{10}$$

and by substituting Equations 8 and 10 into Equation 5, we derive the Diffusion Evolution algorithm: an evolutionary optimization procedure based on iterative error correction akin to diffusion models but without relying on neural networks at all, unlike previous works (Krishnamoorthy et al., 2023; Yan & Jin, 2024). See the psuedocode of Diffusion Evolution in Algorithm 1. When inversely denoising, i.e., evolving from time $T$ to 0, while increasing $\alpha_t$, the Gaussian term will initially have a high variance, allowing global exploration at first. As the evolution progresses, the variance decreases giving lower weight to distant populations, leads to local optimization (exploitation). This locality avoids global competition, akin to reproductive isolation in biology, enabling the algorithm to maintain multiple solutions and balance exploration and exploitation. The weighted average in Equation 7 then acts as a gene recombination operation. Hence, the denoising process of diffusion models can be understood in an evolutionary manner: $\hat{\boldsymbol{x}}_0$ represents an estimated high fitness parameter target. In contrast, $\boldsymbol{x}_t$ can be considered as diffused from high-fitness points. The first two parts in the Equation 5, i.e., $\sqrt{\alpha_{t-1}}\hat{\boldsymbol{x}}_0 + \sqrt{1 - \alpha_{t-1} - \sigma_t^2}\hat{\boldsymbol{\epsilon}}$, guide the individuals towards high fitness targets in small steps. The last part of Equation 5, $\sigma_t \boldsymbol{w}$, is an integral part of diffusion models, perturbing the parameters in our approach similarly to random mutations.

---

**Algorithm 1** Diffusion Evolution

**Require:** Population size $N$, parameter dimension $D$, fitness function $f$, density mapping function $g$, total evolution steps $T$, diffusion schedule $\boldsymbol{\alpha}$ and noise schedule $\boldsymbol{\sigma}$.
**Ensure:** $\alpha_0 \sim 1, \alpha_T \sim 0, \alpha_i > \alpha_{i+1}, 0 < \sigma_i < \sqrt{1 - \alpha_{i-1}}$
1: $[\boldsymbol{x}_T^{(1)}, \boldsymbol{x}_T^{(2)}, ..., \boldsymbol{x}_T^{(N)}] \leftarrow \mathcal{N}(0, I^{N \times D})$                ▷ Initialize population
2: **for** $t \in [T, T-1, ..., 2]$ **do**
3:     $\forall i \in [1, N] : Q_i \leftarrow g[f(\boldsymbol{x}_t^{(i)})]$        ▷ Fitness are cached to avoid repeated evaluations
4:     **for** $i \in [1, 2, .., N]$ **do**
5:         $Z \leftarrow \sum_{j=1}^{N} Q_j \mathcal{N}(\boldsymbol{x}_t^{(i)}; \sqrt{\alpha_t}\boldsymbol{x}_t^{(j)}, 1 - \alpha_t)$
6:         $\hat{\boldsymbol{x}}_0 \leftarrow \frac{1}{Z} \sum_{j=1}^{N} Q_j \mathcal{N}(\boldsymbol{x}_t^{(i)}; \sqrt{\alpha_t}\boldsymbol{x}_t^{(j)}, 1 - \alpha_t)\boldsymbol{x}_t^{(j)}$
7:         $\boldsymbol{w} \leftarrow \mathcal{N}(0, I^D)$
8:         $\boldsymbol{x}_{t-1}^{(i)} \leftarrow \sqrt{\alpha_{t-1}}\hat{\boldsymbol{x}}_0 + \sqrt{1 - \alpha_{t-1} - \sigma_t^2}\frac{\boldsymbol{x}_t^{(i)} - \sqrt{\alpha_t}\hat{\boldsymbol{x}}_0}{\sqrt{1 - \alpha_t}} + \sigma_t \boldsymbol{w}$
9:     **end for**
10: **end for**

---

Figure 2(a) demonstrates the detailed evolution process of a multi-target fitness landscape with two optimal points (see exact fitness function in Appendix A.1). Each individual estimates high fitness parameter targets and moves toward the target along with random mutations. The high fitness parameter targets $\hat{\boldsymbol{x}}_0$ are estimated based on their neighbors' fitness scores (neighbors are shown in blue disks, with radius proportional to $\sqrt{1 - \alpha_t}/\sqrt{\alpha_t}$, see Appendix A.3). The estimated targets $\hat{\boldsymbol{x}}_0$ typically moving toward targets faster than the individuals while the individuals are successively refined in small denoising steps in the direction of the estimated target, see Figure 2(b). Although $\hat{\boldsymbol{x}}_0$ often have higher fitness, they exhibit lower diversity, hence they are used as a goal of individuals instead of the final solutions. This difference also provides flexibility in balancing between more greedy and more diverse strategies.

## 4 EXPERIMENTS

We conduct two sets of experiments to study Diffusion Evolution in terms of diversity and solving complex reinforcement learning tasks. Moreover, we utilize techniques from the diffusion models literature to improve Diffusion Evolution. In the first experiment, we adopt an accelerated sampling method (Nichol & Dhariwal, 2021) to significantly reduce the number of iterations. In the second experiment, we propose Latent Space Diffusion Evolution, inspired by latent space diffusion models (Rombach et al., 2022), allowing us to deploy our approach to complex problems with high-dimensional parameter spaces by exploring a lower-dimensional latent space.

### 4.1 MULTI-TARGET EVOLUTION

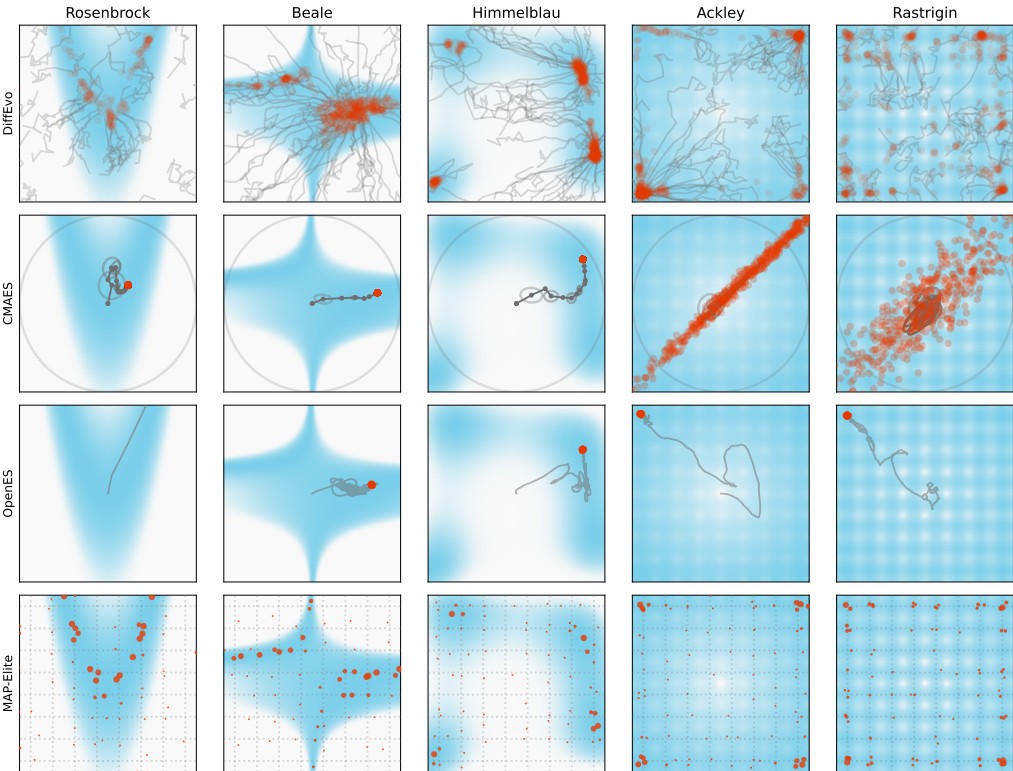

Figure 3: Benchmark experiments (columns) on various fitness functions with selected evolutionary algorithms (rows). Blue (high fitness) and white (low fitness) regions represent a two-dimensional parameter space, with fitness normalized to $[0, 1]$ for comparability (see Appendix). The Diffusion Evolution algorithm identifies multiple optima in 2D benchmarks while preserving genetic diversity. Red dots mark the final population, and gray lines show trajectories for 64 individuals. In CMA-ES, gray ellipsoids represent covariance estimates, and gray lines trace the history of estimated averages.

To compare our method to selected mainstream evolutionary algorithms, we choose five different two-dimensional fitness landscapes as benchmarks: The Rosenbrock and Beale fitness functions have a single optimal point, while the Himmelblau, Ackley, and Rastrigin functions have multiple optimal solutions, see Appendix A.4 for more details; we compare our method to other evolutionary strategies, including CMA-ES (Hansen et al., 2003), OpenES (Salimans et al., 2017), PEPG (Sehnke et al., 2010), and MAP-Elite (Mouret & Clune, 2015). The experiments show that our Diffusion Evolution algorithm can find diverse solutions on the Himmelblau, Ackley, and Rastrigin functions. In contrast, CMA-ES and OpenES struggle, as they either focus on finding a single solution or get distracted by multiple high-fitness peaks, leading to sub-optimal results. While the MAP-Elite method demonstrates diverse solutions, the corresponding average fitness is lower than that of our

Table 1: Average entropy of the top-64 elite populations with 100 different evolutions, along with average fitness (0 to 1) separated by commas. Higher entropy indicates greater diversity, and higher fitness reflects better solutions. See Table 3 in the Appendix for details.

| | Diffusion Evolution | Latent Diffusion Evolution | CMA-ES | PEPG | OpenES | MAP-Elite |
|---|---|---|---|---|---|---|
| Rosenbrock | 5.86, 0.88 | 5.64, 0.92 | 0.00, **1.00** | 0.85, **1.00** | 1.96, 0.71 | **6.00**, 0.77 |
| Beale | 5.50, 0.96 | 5.11, 0.93 | 0.00, **1.00** | 0.35, **1.00** | 1.03, **1.00** | **6.00**, 0.37 |
| Himmelblau | 5.23, 0.96 | 4.95, 0.89 | 0.00, **1.00** | 0.00, **1.00** | 0.28, **1.00** | **6.00**, 0.28 |
| Ackley | 5.67, 0.78 | 5.31, 0.73 | 5.34, 0.66 | 0.04, 0.96 | 0.18, **1.00** | **5.98**, 0.50 |
| Rastrigin$^2$ | 5.79, 0.64 | 5.34, 0.62 | 5.70, 0.57 | 0.00, 0.57 | 0.01, **1.00** | **5.95**, 0.61 |
| Rastrigin$^4$ | 5.82, 0.18 | **5.99**, **0.37** | 5.67, 0.17 | 0.00, 0.17 | 0.00, 0.25 | 5.95, 0.18 |
| Rastrigin$^{32}$ | 5.84, 0.02 | **6.00**, **0.19** | 4.54, 0.01 | 0.00, 0.02 | 0.00, 0.02 | 5.94, 0.02 |
| Rastrigin$^{256}$ | 5.84, 0.00 | **6.00**, **0.15** | 2.41, 0.00 | 0.00, 0.00 | 1.65, 0.00 | 5.95, 0.00 |

Notations: **highest value**, second highest value, third highest value. Superscript numbers on Rastrigin functions indicate the dimensions of the fitness function.

method. Our experiments demonstrate that the Diffusion Evolution algorithm can identify high-fitness and diverse solutions and adapt to various fitness landscapes (see Figure 3 and Table 1).

The most time-consuming part of evolutionary algorithms is often the fitness evaluation. In this experiment, we adopt an accelerated sampling method from the diffusion models literature to reduce the number of iterations. As proposed by Nichol & Dhariwal (2021), instead of the default $\alpha_t$ scheduling of DDPM, a cosine scheduling $\alpha_t = \cos(\pi t/T)/2 + 1/2$ leads to better performance when $T$ is small (see Appendix A.2 for a detailed comparison). With this, we can significantly reduce the number of fitness evaluations while maintaining sampling diversity and quality.

To systematically compare different methods, we repeated the evolution 100 times for each method. In all experiments, we rescaled the fitness functions to the range 0 to 1, with 1 representing the highest fitness (see Appendix A.4). Each experiment was conducted with a population of 512 for 25 iterations, except for the OpenES method, which requires 1000 steps to converge. To quantify diversity, we then calculated the Shannon entropy of the final population by gridding the space and counting the individuals in different grid cells (we select the top-64 fitness individuals, focusing solely on elite individuals). The results in Table 1 show that our method consistently finds diverse solutions without sacrificing fitness performance. While CMA-ES shows high entropy on the Ackley and Rastrigin functions, it finds significantly lower fitness solutions compared to Diffusion Evolution, suggesting CMA-ES is distracted by multiple solutions rather than finding diverse ones (see examples in Figure 3). Similarly, the MAP-Elite algorithm shows the highest diversity but compromises on fitness, yielding the lowest fitness for most tasks. In contrast, our Diffusion Evolution algorithm neurally excels at balancing quality and diversity, despite not being explicitly designed for this purpose, as detailed in Appendix A.4.3.

## 4.2 LATENT SPACE DIFFUSION EVOLUTION

Here, we apply the Diffusion Evolution method to reinforcement learning tasks (Sutton & Barto, 1998) to train neural networks for controlling the cart-pole system (Barto et al., 1983), among other tasks (see Appendix A.5). This system has a cart with a hinged pole, and the objective is to keep the pole vertical as long as possible by moving the cart sideways while not exceeding a certain range, see Figure 4(d). The game is terminated if the pole angle exceeds $\pm 12°$ or the cart position exceeds $\pm 2.4$. Thus, longer duration yields higher fitness. We use a two-layer neural network of 58 parameters to control the cart, with inputs being the current position, velocity, pole angle, and pole angular velocity. The output of the neural network determines whether to move left or right. See more details about the neural network in Appendix A.5.1. The task is considered solved when a fitness score (cumulative reward) of 500 is reached consistently over several episodes.

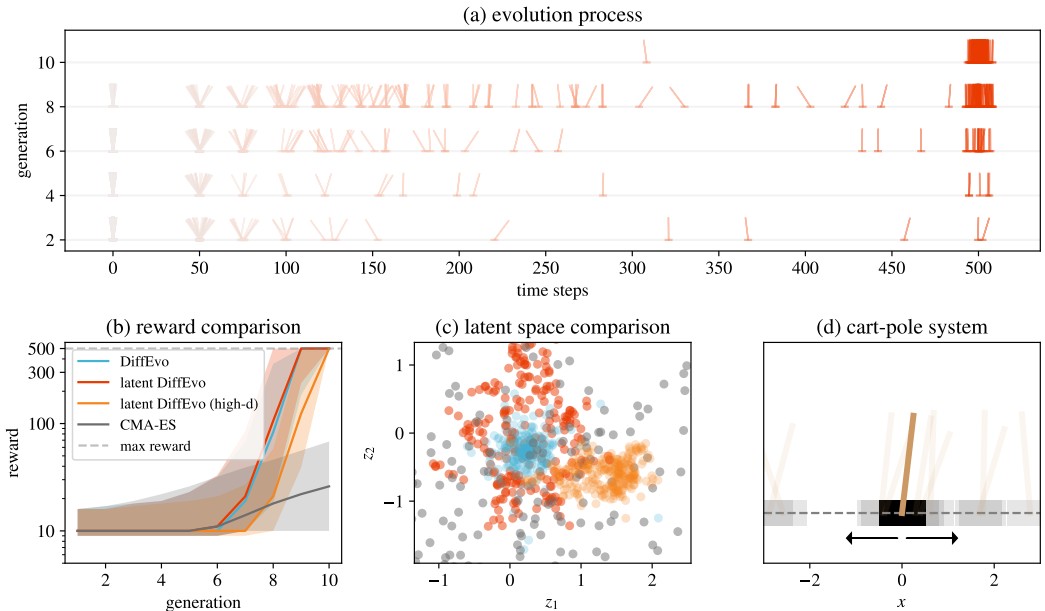

Figure 4: **(a)** Evolution of cart-pole tasks: The horizontal axis shows survival time, and the vertical axis represents generations. Each point represents an individual's final state (pole angle, cart position). Over time, agents survive longer and achieve higher rewards. **(b)** Latent Diffusion Evolution methods (red) find effective solutions within ten steps, whereas CMA-ES (gray) fails within the given generations. This latent approach also works in high-dimensional spaces (orange) with up to 17,410 dimensions. Experiments are repeated 100 times, with medians (solid lines) and ranges (25%–75%) shown as shaded areas. **(c)** Projecting individuals' parameters into a latent space visualizes diversity. The same projection is used for all results except for the high-dimensional case. This highlights the increased solution diversity of the latent method. **(d)** The cart-pole system includes a pole hinged to a cart, where the controller balances the pole by moving the cart left or right.

Deploying our original Diffusion Evolution method to high-dimensional parameter spaces may result in poor performance, see high-dimensional Rastrigin functions in Table 1. To address this issue, we propose *Latent Space Diffusion Evolution*: inspired by the latent space diffusion model (Rombach et al., 2022), we map individual parameters into a lower-dimensional latent space in which we perform the Diffusion Evolution algorithm. This approach significantly improves performance. This approach is motivated from analyzing the Gaussian term in Equation 8 which we use to estimate the original points $\hat{x}_0$. In higher dimensions, the increased pairwise distances between parameters cause the evolution process to become more localized and consequently slower. Moreover, the parameter or genotype space may have dimensions that do not effectively impact fitness, known as sloppiness (Gutenkunst et al., 2007). Assigning random values to these dimensions often does not affect fitness, similar to genetic drift (KIMURA, 1991) or neutral genes (King & Jukes, 1969), suggesting the true high-fitness genotype distribution is lower-dimensional. The straightforward approach is directly denoising in a lower-dimensional latent space $z$ and estimating high-quality targets $z_0$ via:

$$\hat{z}_0(z_t, \alpha, t) = \sum_z p(z|z_t)z = \frac{1}{Z}\sum_{x \sim p_{\text{eval}}(x)} g[f'(z)]\mathcal{N}(z_t; \sqrt{\alpha_t}z, 1 - \alpha_t)z. \quad (11)$$

However, this approach requires a decoder and a new fitness function $f'$ for $z$, which can be challenging to obtain. To circumvent this, we approximate latent diffusion by using the latent space only to calculate the distance between individuals. Although we do not know the exact distribution of $x$ a priori, a random projection can often preserve distance relations between populations, as suggested by the Johnson-Lindenstrauss lemma (Johnson, 1984). To this end, we change Equation 11 to:

$$\hat{x}_0(x_t, \alpha, t) = \sum_{x \sim p_{\text{eval}}(x)} p(x|x_t)x \approx \frac{1}{Z}\sum_{x \sim p_{\text{eval}}(x)} g[f(x)]\mathcal{N}(z_t; \sqrt{\alpha_t}z, 1 - \alpha_t)x, \quad (12)$$

where $z = Ex$, $E_{ij} \sim \mathcal{N}^{(d,D)}(0, 1/D)$, $D$ is the dimension of $x$, and $d$ is the latent space dimension (Johnson, 1984), see Algorithm 2 in the Appendix. Here we choose $d = 2$ in our experiments for visualization purposes. The results show the effectiveness of Latent Diffusion Evolution on the cart-pole task and significant improvements in benchmarks and other reinforcement learning tasks (see Table 1 and 5). We found that latent evolution remains effective even within a significantly higher-dimensional parameter space. For example, using a three-layer neural network with $17,410$ parameters, it still achieving strong performance, see Figure 4(b). By combining this approach with an accelerated sampling method, the cart-pole task can be solved in just 10 generations, using a population size of $512$ and one fitness evaluation per individual. As shown in Appendix A.5, our approach demonstrates effectiveness in a variety of reinforcement learning tasks. This highlights the potential of using tools and theories from the diffusion model domain in evolutionary optimization tasks and *vice versa*, opening up broad opportunities to improve and understand evolution from a new perspective.

## 5 DISCUSSION

By aligning diffusion models with evolutionary processes, we demonstrate that *diffusion models are evolutionary algorithms, and evolution can be viewed as a generative process*. The Diffusion Evolution process inherently includes mutation, selection, hybridization, and reproductive isolation, indicating that diffusion and evolution are *two sides of the same coin*. Our Diffusion Evolution algorithm leverages this theoretical connection to improve solution diversity without compromise quality too much compared to standard approaches. By integrating latent space diffusion and accelerated sampling, our method scales to high-dimensional spaces, enabling the training of neural network agents in reinforcement learning environments with exceptionally short training time.

This equivalence between the two fields offers valuable insights from both deep learning and evolutionary computation. Through the lens of machine learning, the evolutionary process can be viewed as nature's way of learning and optimizing strategies for survival of species over generations. Similarly, our Diffusion Evolution algorithm iteratively refines estimation of high-fitness parameters, continuously learning and adapting to the fitness landscape. This positions evolutionary algorithms not merely as optimization tools, but also as learning frameworks that enhance understanding and functionality through iterative refinement. Conversely, framing evolution as a diffusion process offers a concrete mathematical formulation. In contrast to previous work (Ao, 2005), we provide an explicit and implementable evolutionary framework.

The connection between diffusion and evolution enables mutual contributions between the two fields. Diffusion models are extensively studied in the contexts of controlling, optimization, and probability theory, offering robust tools to analyze and enhance evolutionary algorithms. In our experiments, leveraging concepts from diffusion models enabled flexible strategies while maintaining the effectiveness of evolutionary processes. For instance, accelerated sampling methods (Nichol & Dhariwal, 2021) can be applied seamlessly to Diffusion Evolution to accelerate the optimization process. Latent diffusion models (Rombach et al., 2022) inspired our Latent Space Diffusion Evolution, enabling evolution in high-dimensional spaces and substantially improving performance. Other advancements in the diffusion model field hold the potential to enhance our understanding of evolutionary processes. For instance, non-Gaussian noise diffusion models (Bansal et al., 2024), discrete denoising diffusion models, classifier free guidance (Ho & Salimans, 2022), and theoretical studies, e.g., spontaneous symmetry breaking (Raya & Ambrogioni, 2024) in the generative process of diffusion models unveil entirely new possibilities and perspectives for understanding and advancing evolutionary methods; c.f. our complimentary Conditional Diffusion Evolution (Hartl et al., 2024). While diffusion models are inherently designed with a finite number of sampling steps, viewing them through the lens of non-equilibrium thermodynamics, such as Langevin dynamics (Song et al., 2020b), may allow for open-ended evolution, a characteristic inherent in natural evolution.

However, this parallel we draw here between evolution and diffusion models also gives rise to several challenges and open questions. Could other diffusion model implementations yield different evolutionary methods with diverse and unique features? Can advancements in diffusion models help introduce inductive biases into evolutionary algorithms? How do latent diffusion models correlate with neutral genes? Additionally, can insights from the field of evolution enhance diffusion models? These questions highlight the potential of this duality and synergy between diffusion and evolution.

ACKNOWLEDGMENTS

The authors acknowledge the Tufts University High Performance Compute Cluster [2] and the Vienna Scientific Cluster [3] which have been utilized for the research reported in this paper. M.L. gratefully acknowledges support via grant 62212 from the John Templeton Foundation, via grant TWCF0606 from the Templeton World Charity Foundation, and via a sponsored research agreement from Astonishing Labs. BH acknowledges an APART-MINT stipend by the Austrian Academy of Sciences.

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

## A APPENDIX

### A.1 TWO-PEAKS MODELS

We first apply our method to a simple two-dimensional fitness function, with two optimal points, to demonstrate its behavior and capability of finding multiple solutions. We choose a continuous mixed Gaussian density function. The fitness function is a mixed Gaussian density function with means located at $(1, 1)$ and $(-1, -1)$. The fitness function is:

$$f(x, y) = \left[ \mathcal{N}((x, y); \boldsymbol{\mu}_1, \sigma^2) + \mathcal{N}((x, y); \boldsymbol{\mu}_2, \sigma^2) \right] / 2, \tag{13}$$

where $\boldsymbol{\mu}_1 = (1, 1)$ and $\boldsymbol{\mu}_2 = (-1, -1)$. And the $\sigma = 0.1$.

### A.2 ALPHA AND NOISE SCHEDULE

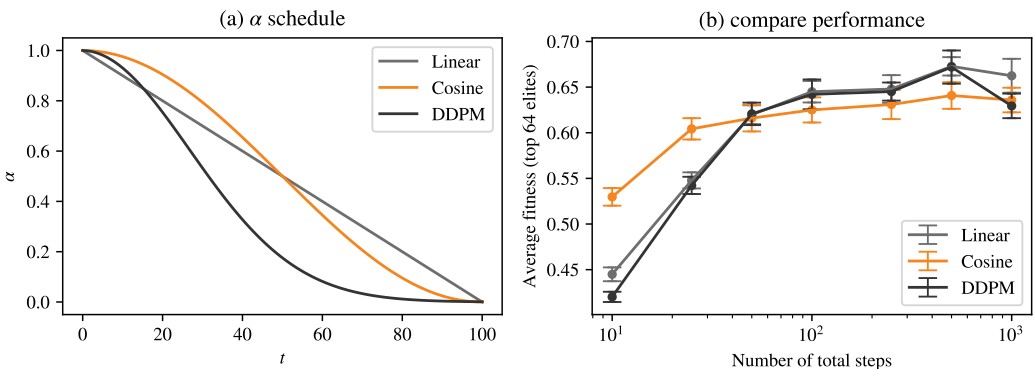

Figure 5: **(a)** Different $\alpha$ scheduling methods used, here for a total of $T = 100$ evolutionary steps. **(b)** Average final fitness and standard deviation of 100 experiments of 2-dimensional Rastrigin function with different numbers of total evolutionary steps is analyzed under various $\alpha$ scheduling methods. In general, a larger number of total evolution steps lead to higher average fitness. Most $\alpha$ scheduling methods result in fitness plateauing around $T = 100$. However, cosine scheduling achieves higher fitness when the total number of evolution steps is lower than 100, demonstrating its capability to accelerate sampling by reducing the total number of fitness evaluations.

In our experiments, we tested three different schedules for $\alpha_t$, see Figure 5(a). The first is a simple linear schedule, used in the two-peaks demonstration:

$$\alpha_t = 1 - \frac{t}{T}. \tag{14}$$

The second is the schedule used in DDPM (Ho et al., 2020), which can be approximated by:

$$\alpha_t = \exp\left(-\beta_0 t - \frac{\gamma t^2}{T}\right), \tag{15}$$

where $\beta_0$ and $\gamma$ are hyperparameters. These are calculated by constraining $\alpha_0 = 1 - \varepsilon$ and $\alpha_T = \varepsilon$, with $\varepsilon = 10^{-4}$ as the default.

The third schedule is the cosine schedule proposed by Nichol & Dhariwal (2021) and is used in both Figures 3 and 4:

$$\alpha_t = \frac{1}{2} \cos\left(\frac{\pi t}{T}\right) + \frac{1}{2}. \tag{16}$$

The cosine $\alpha$ scheduling demonstrates better performance when the total number of evolution steps are fewer than 100 (see Figure 5(b)), aligning with the conclusions reported by Nichol & Dhariwal (2021). Hence, we use cosine $\alpha$ scheduling by default unless otherwise specified.

For $\sigma_t$, we follow the DDIM setting with a slight modification for better control:

$$\sigma_t = \sigma_m \sqrt{\frac{1 - \alpha_{t-1}}{1 - \alpha_t}} \sqrt{1 - \frac{\alpha_t}{\alpha_{t-1}}}, \qquad (17)$$

where $0 \leq \sigma_m \leq 1$ is a hyperparameter to control the magnitude of noise. We use $\sigma_m = 1$ for most experiments, except for the experiment demonstrating the process in Figure 2, which requires a lower noise magnitude $\sigma_m = 0.1$ for better visualization.

## A.3   NEIGHBORHOOD OF INDIVIDUALS

In Figure 2, we use blue discs to represent the neighborhood of each individual. The mean and standard deviation of an individual's neighborhood can be derived from the Gaussian term in Equation 8. Transforming this term into an equivalent form with $\boldsymbol{x}$ as the variable and $\boldsymbol{x}_t$ as the parameter yields:

$$\mathcal{N}(\boldsymbol{x}_t; \sqrt{\alpha_t}\boldsymbol{x}, 1 - \alpha_t) = \frac{1}{\sqrt{\alpha_t}}\mathcal{N}\left(\boldsymbol{x}; \frac{\boldsymbol{x}_t}{\sqrt{\alpha_t}}, \sqrt{\frac{1 - \alpha_t}{\alpha_t}}\right). \qquad (18)$$

Hence, this Gaussian term can be transformed into a form where $\boldsymbol{x}$ is the variable, which is more intuitive for identifying data points $\boldsymbol{x}$ as the neighbors of $\boldsymbol{x}_t/\sqrt{\alpha_t}$. Thus, the neighborhood-discs are centered at $\boldsymbol{\mu} = \boldsymbol{x}_t/\sqrt{\alpha_t}$ and have a squared radius of $r^2 = (1 - \alpha_t)/\alpha_t$.

## A.4   FITNESS FUNCTIONS

Table 2: Fitness functions used in our experiments. The superscript on Rastrigin indicates the dimension of the function.

| Name | Formula | features |
|---|---|---|
| Rosenbrock | $f(x, y) = 100(y - x^2)^2 + (1 - x)^2$ | The minimal value position is $(x, y) = (1, 1)$, where $f(1, 1) = 0$. |
| Beale | $f(x, y) = (1.5 - x + xy)^2 + (2.25 - x + xy^2)^2 + (2.625 - x + xy^3)^2$ | The minimal value position is $(x, y) = (3, 0.5)$, where $f(3, 0.5) = 0$. |
| Himmelblau | $f(x, y) = (x^2 + y - 11)^2 + (x + y^2 - 7)^2$ | This function has four minimal value points, they are: $f(3.0,\ 2.0) = 0.0$, $f(-2.81, 3.13) = 0.0$, $f(-3.78, -3.28) = 0.0$, $f(3.58, -1.85) = 0.0$ |
| Ackley | $f(x, y) = -20 \exp\left(-0.2\sqrt{\frac{x^2 + y^2}{2}}\right) - \exp\left(\frac{\cos 2\pi x + \cos 2\pi y}{2}\right) + e + 20$ | When restricting the range of $x, y$ between -4 to 4, the maximal points are located at the four corners. |
| Rastrigin$^n$ | $f(\boldsymbol{x}) = An + \sum_{i=1}^{n}[\boldsymbol{x}_i^2 - A\cos(2\pi\boldsymbol{x}_i)]$ | Here $A = 10$, and $n$ is the dimension. Similar as the Ackley function above, when restricting the range of $\boldsymbol{x}$, the maximal points are located at the four corners. |

To benchmark the solution diversity and performance, we choose eight different fitness functions to compare our method with other evolutionary strategies. All the functions depend on variable $\boldsymbol{x}$,

with the objective being to minimize or maximize the function value. Specifically, we constrain the range of $\boldsymbol{x}_i$ to $(-4, 4)$ and set the objective of the Rastrigin function to be maximization instead of minimization to benchmark the capability of finding multiple solutions. All the functions are scaled by 4, i.e., $f(\boldsymbol{x}) \to f'(4\boldsymbol{x})$, to ensure that the standard Gaussian distribution can cover the parameter space. The details of these fitness functions are shown in Table 2.

### A.4.1 PROBABILITY MAPPING FUNCTION

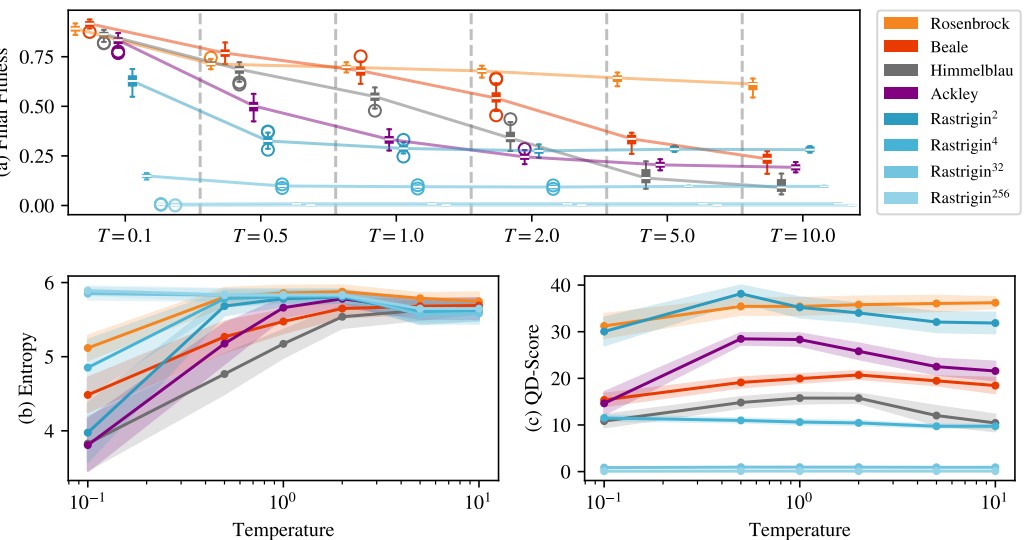

Figure 6: Effects of temperature $T$ on the probability mapping function $g$. **(a)** Average fitness and standard deviation of 100 run with $g$ functions under different temperature $T$ settings are shown. A value of one represents the highest fitness, while zero indicates the lowest fitness. The general trend reveals that as $T$ increases, Diffusion Evolution tends to sample fewer high-fitness populations. **(b)** The average entropy of sampled parameters increases as temperature rises, plateauing rapidly around $T = 1$. The shaded areas indicate the standard deviation of the entropy. **(c)** When measured by QD-score, many tasks achieve their maximum QD-score around $T = 1$, suggesting this is the optimal setting for most tasks.

To standardize the comparison, we apply the following fitness mapping function $g$ to convert target values to the highest fitness:

$$(g \circ f)(\boldsymbol{x}) = \frac{e^{-D(f(\boldsymbol{x}), f^*)/(sT)} - g_{\min}}{1 - g_{\min}} \tag{19}$$

Here, $T$ is the temperature parameter used to control the sharpness of the probability density function, and $f^*$ is the target fitness value, with $s$ as the scale factor to make different functions comparable. The $D$ represents the distance function, defined as $D(x, y) = |x - y|$, and $D_{\max}$ is the maximum possible distance between the target fitness and the sampled fitness within the given parameter region. The $g_{\min}$ is defined as $\exp(-D_{\max}/(sT))$, representing the minimal possible probability value before rescaling. For each fitness function, we determine $f^*$ as the minimal or maximal value, depending on the optimization objective. The scale factor is determined by the standard deviation of fitness around their optimal points, with ranges adjusted for different functions. After this transformation, the lowest fitness is zero, and the highest fitness is one.

While we acknowledge that the choice of the probability mapping function can vary depending on the task, we use Equation 19 with a $T$ hyper-parameter to investigate how the probability mapping function affects the results and to determine the optimal $g$ function. As illustrated in Figure 6, higher temperatures generally make fitness values less distinguishable, leading the algorithm to sample fewer high-fitness individuals while simultaneously increasing diversity. By evaluating the

QD-score, as shown in Figure 6(c), we recommend selecting $T = 1$ for most probability mapping functions when an appropriate scale factor $s$ is applied. For this reason, we use $T = 1$ for all benchmark and reinforcement learning experiments, except in Figure 3, where we set $T = 0.25$ for better visualization.

Table 3: A detailed version of Table 1 is presented, where each cell contains two lines. The first line indicates the entropy of the top-64 elite populations after evolution, while the second line represents the average fitness (ranging from 0 to 1). Standard deviations for both entropy and fitness are provided in parentheses. The superscript numbers on the Rastrigin functions indicate the dimension of the fitness function.

| | Diffusion Evolution | Latent Diffusion Evolution | CMA-ES | PEPG | OpenES | MAP-Elite |
|---|---|---|---|---|---|---|
| Rosenbrock | 5.86 (0.10) | 5.64 (0.24) | 0.00 (0.00) | 0.85 (0.26) | 1.96 (0.21) | **6.00** (0.01) |
| | 0.88 (0.06) | 0.92 (0.06) | **1.00** (0.00) | **1.00** (0.00) | 0.71 (0.00) | 0.77 (0.00) |
| Beale | 5.50 (0.16) | 5.11 (0.37) | 0.00 (0.00) | 0.35 (0.21) | 1.03 (0.45) | **6.00** (0.01) |
| | 0.96 (0.01) | 0.93 (0.06) | **1.00** (0.00) | **1.00** (0.00) | **1.00** (0.01) | 0.37 (0.01) |
| Himmelblau | 5.23 (0.20) | 4.95 (0.39) | 0.00 (0.00) | 0.00 (0.00) | 0.28 (0.46) | **6.00** (0.01) |
| | 0.96 (0.01) | 0.89 (0.11) | **1.00** (0.00) | **1.00** (0.00) | **1.00** (0.00) | 0.28 (0.01) |
| Ackley | 5.67 (0.13) | 5.31 (0.42) | 5.34 (1.30) | 0.04 (0.17) | 0.18 (0.38) | **5.98** (0.02) |
| | 0.78 (0.07) | 0.73 (0.17) | 0.66 (0.20) | 0.96 (0.13) | **1.00** (0.00) | 0.50 (0.01) |
| Rastrigin$^2$ | 5.79 (0.10) | 5.34 (0.43) | 5.70 (0.38) | 0.00 (0.00) | 0.01 (0.04) | **5.95** (0.04) |
| | 0.64 (0.04) | 0.62 (0.08) | 0.57 (0.11) | 0.57 (0.16) | **1.00** (0.00) | 0.61 (0.01) |
| Rastrigin$^4$ | 5.82 (0.09) | **5.99** (0.01) | 5.67 (0.59) | 0.00 (0.00) | 0.00 (0.00) | 5.95 (0.04) |
| | 0.18 (0.01) | **0.37** (0.03) | 0.17 (0.02) | 0.17 (0.03) | 0.25 (0.00) | 0.18 (0.00) |
| Rastrigin$^{32}$ | 5.84 (0.07) | **6.00** (0.00) | 4.54 (1.13) | 0.00 (0.00) | 0.00 (0.00) | 5.94 (0.04) |
| | 0.02 (0.00) | **0.19** (0.01) | 0.01 (0.00) | 0.02 (0.00) | 0.02 (0.00) | 0.02 (0.00) |
| Rastrigin$^{256}$ | 5.84 (0.08) | **6.00** (0.00) | 2.41 (1.78) | 0.00 (0.00) | 1.65 (0.30) | 5.95 (0.04) |
| | 0.00 (0.00) | **0.15** (0.00) | 0.00 (0.00) | 0.00 (0.00) | 0.00 (0.00) | 0.00 (0.00) |

Notations: **highest value**, second highest value, third highest value.

Superscript numbers on Rastrigin functions indicate the dimensions of the fitness function.

### A.4.2   ESTIMATING ENTROPY TO QUANTIFY DIVERSITY

To quantify the diversity of the solutions, we divided the $D$-dimensional space into $80^D$ grids and counted the frequencies of elite solutions within this space. We intentionally used this simple and coarse method to quantify entropy in order to eliminate the contribution of local diversity, focusing solely on the diversity of solutions across different basins. The entropy is calculated by:

$$H = \sum_{i=1}^{N} P_i \log_2 P_i, \tag{20}$$

where $P_i$ is the probability of having a sample in grid $i$.

### A.4.3   QUALITY-DIVERSITY SEARCH

We use MAP-Elite (Mouret & Clune, 2015) to perform quality-diversity search experiments, enabling us to compare our method's ability to balance quality and diversity. Briefly, in addition to the fitness function, MAP-Elite requires an additional feature descriptor, $c(\boldsymbol{x})$. Using this feature descriptor, MAP-Elite only compares individuals within the same feature class. If a new individual introduces a novel feature, it is accepted and stored regardless of its fitness. Conversely, if the feature is already present, the fitness of the new individual is compared to the existing best individual

within that feature class. To ensure a fair comparison between MAP-Elite and other evolutionary tasks, we use the same number of fitness evaluations. For instance, if Diffusion Evolution uses 512 populations and 25 iterations, then MAP-Elite is allocated $512 \times 25 = 12,800$ iterations, with each iteration proposing one new individual and evaluating its fitness once.

To apply MAP-Elite to benchmark functions, we manually designed a feature descriptor by dividing the parameter space into segments of a specified unit length $l$. Specifically, the feature descriptor is defined as:

$$c(\boldsymbol{x}) = (\lfloor \boldsymbol{x}_1/l \rfloor, \lfloor \boldsymbol{x}_2/l \rfloor, \cdots, \lfloor \boldsymbol{x}_D/l \rfloor). \tag{21}$$

For simplicity, we set $l = 1$ in all experiments. We quantify the balance between quality and diversity using the quality-diversity score (QD-score) (Pugh et al., 2016), calculated as the sum of the highest rewards in each feature class. For methods other than MAP-Elite, we apply the same classification approach to divide individuals into feature classes and compute the QD-score by summing the highest rewards within each class. The comparison of QD-scores is presented in Table 4.

In two-dimensional fitness functions, Diffusion Evolution and Latent Diffusion Evolution achieve QD-scores comparable to MAP-Elite. However, in higher-dimensional tasks, the QD-score of MAP-Elite drops significantly, whereas our Latent Diffusion Evolution demonstrates substantial better performance. This result indicates that our method effectively balances quality and diversity, particularly in high-dimensional settings, without being designed to do so.

Table 4: The QD-scores of experiments on fitness functions with different algorithms are reported. All values are averaged over 100 experiments, with the standard deviations of the QD-scores provided in brackets. Our Diffusion Evolution demonstrates competitive performance on QD-scores compared to MAP-Elite. In higher-dimensional experiments, our Latent Diffusion Evolution achieves the highest QD-score among all tested methods.

| | Diffusion Evolution | Latent Diffusion Evolution | CMA-ES | PEPG | Open-ES | MAP-Elite |
|---|---|---|---|---|---|---|
| Rosenbrock | 35.4 (1.82) | 23.4 (9.78) | 1.00 (0.00) | 1.25 (0.43) | 0.73 (0.12) | **42.0** (0.36) |
| Beale | 20.0 (0.94) | 13.0 (4.50) | 1.00 (0.00) | 2.00 (0.00) | 1.84 (0.72) | **23.7** (0.48) |
| Himmelblau | 15.8 (1.18) | 11.4 (3.99) | 1.00 (0.00) | 1.00 (0.00) | 1.00 (0.00) | **18.1** (0.42) |
| Ackley | 28.3 (1.35) | 16.7 (9.07) | 13.4 (4.53) | 1.13 (0.34) | 2.14 (0.63) | **33.0** (0.53) |
| Rastrigin$^2$ | 35.2 (2.18) | 17.8 (9.36) | 18.9 (6.24) | 2.30 (0.64) | 3.99 (0.00) | **45.2** (0.93) |
| Rastrigin$^4$ | 10.6 (0.50) | **33.1** (6.66) | 6.03 (2.30) | 0.67 (0.12) | 1.01 (0.00) | 13.5 (0.22) |
| Rastrigin$^{32}$ | 0.96 (0.04) | **73.4** (2.10) | 0.12 (0.03) | 0.06 (0.01) | 0.07 (0.01) | 1.20 (0.02) |
| Rastrigin$^{256}$ | 0.11 (0.01) | **70.2** (0.62) | 0.01 (0.00) | 0.01 (0.00) | 0.00 (0.00) | 0.14 (0.00) |

Notations: **highest value**, second highest value, third highest value.

Superscript numbers on Rastrigin functions indicate the dimensions of the fitness function.

## A.5 REINFORCEMENT LEARNING EXPERIMENTS

We applied our Diffusion Evolution and Latent Diffusion Evolution methods to various reinforcement learning tasks using neural network architectures described in Appendix A.5.1, including Acrobot, Cart Pole, Mountain Car, Continuous Mountain Car, and Pendulum. Our approach consistently achieves higher final rewards compared to CMA-ES, with statistically significant differences.

To assess the robustness of our method, we introduced a new scaling hyper-parameter $s$, which transforms the original fitness function $f(x)$ into $f_s(x) = f(sx)$. This enables evaluation of performance across diverse parameter spaces. The results, presented in Table 5 and 6, demonstrate that our method is robust across different choices of the scaling parameter $s$. Furthermore, we observe that Latent Diffusion Evolution is always outperforming the original Diffusion Evolution in terms of success rate, supporting the rationale for introducing the latent approach. The CMA-ES method,

although successful with certain populations, does not achieve a high success rate within the given steps. We hypothesize that the reason might be that CMA-ES is distracted by multiple solutions, leading to a significant proportion of unsuccessful outcomes.

Table 5: Grid search of reinforcement learning tasks with multiple environments, incorporating a parameter scaling factor, was conducted. This scaling factor tests different landscapes of the parameter space and evaluates the robustness of evolutionary algorithms. Here, higher rewards indicate better performance, with the highest rewards per line shown in **bold**, and standard deviations (over 100 experiments) presented in brackets. The results demonstrate that Diffusion Evolution algorithms consistently achieve the highest rewards across different scaling factors. Furthermore, Latent Diffusion Evolution exhibits greater stability with respect to the scaling factor and often yields higher rewards.

| Scaling | Environment | Diffusion Evolution | Latent Diffusion Evolution | Large Latent Diffusion Evolution | CMA-ES |
|---|---|---|---|---|---|
| 0.1 | Acrobot | -279.5 (187.2) | **-149.0** (115.0) | -157.6 (116.9) | -486.9 (56.7) |
| | Cart Pole | **447.4** (121.2) | 447.0 (124.3) | 407.6 (154.9) | 32.64 (71.81) |
| | Mountain Car | -163.2 (39.39) | -139.7 (35.63) | **-138.1** (37.58) | -199.2 (7.29) |
| | Mountain Car$^\dagger$ | -0.89 (1.19) | -0.02 (0.05) | **-0.01** (0.05) | -0.14 (0.21) |
| | Pendulum | -1231 (337.6) | **-1224** (347.0) | -1227 (340.7) | -1257 (302.6) |
| 1.0 | Acrobot | -199.9 (160.3) | **-127.0** (93.06) | -147.6 (107.2) | -471.0 (81.48) |
| | Cart Pole | 482.9 (73.24) | **491.6** (50.2) | 445.0 (124.20) | 77.67 (127.3) |
| | Mountain Car | -134.7 (34.8) | **-130.6** (33.3) | -134.8 (37.5) | -194.7 (18.3) |
| | Mountain Car$^\dagger$ | 55.97 (47.9) | 78.59 (39.05) | **88.58** (21.66) | 33.94 (63.75) |
| | Pendulum | -1262 (330.4) | -1187 (408.5) | **-1094** (532.6) | -1397 (217.4) |
| 10.0 | Acrobot | -191.8 (156.3) | **-121.0** (76.98) | -149.8 (105.0) | -469.2 (83.56) |
| | Cart Pole | 478.7 (78.58) | **488.6** (59.73) | 428.9 (142.7) | 79.47 (130.3) |
| | Mountain Car | -134.2 (34.9) | **-129.5** (32.78) | -133.9 (36.60) | -194.85 (17.89) |
| | Mountain Car$^\dagger$ | 79.44 (37.46) | **91.66** (11.30) | 83.41 (33.82) | 10.90 (68.52) |
| | Pendulum | -1132 (488.4) | **-1077** (520.5) | -1101 (519.5) | -1368 (246.34) |
| 100.0 | Acrobot | -190.7 (156.28) | **-120.6** (76.73) | -151.7 (110.32) | -469.6 (83.27) |
| | Cart Pole | 477.9 (82.10) | **489.5** (57.86) | 443.1 (128.48) | 77.72 (127.89) |
| | Mountain Car | -133.6 (34.99) | **-130.6** (33.74) | -134.6 (37.45) | -194.6 (18.22) |
| | Mountain Car$^\dagger$ | 78.56 (39.49) | **90.67** (16.02) | 82.38 (35.57) | 12.97 (69.27) |
| | Pendulum | -1119 (494.4) | **-1066** (535.6) | -1102 (521.7) | -1367 (243.0) |

Mountain Car$^\dagger$: continuous version of Mountain Car
Notations: **highest value**, second highest value, third highest value.

### A.5.1 NEURAL NETWORK

The controller of the cart-pole system has four observational inputs: the current position, velocity, pole angle, and pole angular velocity. The system accepts two actions: move left or right. To model the controller, we use artificial neural networks with an input layer of 4 neurons corresponding to the four observations and an output layer of 2 neurons corresponding to the two actions. The action is determined by which output neuron has the higher value.

Our standard experiment uses a one-hidden-layer neural network with the hidden layer of 8 neurons, resulting in $(4 \times 8 + 8) + (8 \times 2 + 2) = 58$ parameters. We also use a deeper neural network with two hidden layers (each has 128 neurons), totaling $(4 \times 128 + 128) + (128 \times 128 + 128) + (128 \times 2 + 2) = 17,410$ parameters. Both neural networks use the ReLU activation function.

We use similar neural network configurations for other reinforcement learning tasks, with adaptations based on the number of observations as well as the size and type (continuous or discrete) of

Table 6: Success rate of selected environments with different scaling factors. The Pendulum experiment is not included as it does not have a definition of success.

| Scaling | Environment | Diffusion Evolution | Latent Diffusion Evolution | Large Latent Diffusion Evolution | CMA-ES |
|---------|-------------|---------------------|----------------------------|----------------------------------|--------|
| 0.1 | Acrobot | 59.3% | **91.6%** | 91.1% | 6.2% |
| | Cart Pole | 80.0% | **80.8%** | 68.7% | 1.5% |
| | Mountain Car | 53.8% | **86.6%** | 88.6% | 1.6% |
| | Mountain Car† | 0.0% | 0.0% | 0.0% | 0.0% |
| 1.0 | Acrobot | 79.1% | **95.2%** | 93.0% | 13.5% |
| | Cart Pole | 92.5% | **96.4%** | 79.4% | 5.9% |
| | Mountain Car | 92.0% | **97.1%** | 91.9% | 10.2% |
| | Mountain Car† | 59.4% | 80.7% | **96.9%** | 56.7% |
| 10.0 | Acrobot | 80.9% | **97.3%** | 93.6% | 14.4% |
| | Cart Pole | 90.4% | **95.6%** | 76.4% | 6.4% |
| | Mountain Car | 93.1% | **98.2%** | 93.7% | 10.2% |
| | Mountain Car† | 92.0% | **99.3%** | 94.0% | 43.5% |
| 100.0 | Acrobot | 81.0% | **97.2%** | 92.6% | 14.1% |
| | Cart Pole | 90.9% | **95.8%** | 80.0% | 6.0% |
| | Mountain Car | 93.5% | **97.1%** | 92.1% | 10.5% |
| | Mountain Car† | 91.4% | **98.7%** | 93.3% | 45.3% |

Mountain Car†: continuous version of Mountain Car
Notations: **highest value**, second highest value, third highest value.

operations. The design of neural networks generally follows similar principles in machine learning. Specifically, we rescaled the input to have a standard deviation of one to improve training.

## A.6 LATENT SPACE DIFFUSION EVOLUTION

Following is the pseudocode for Latent Space Diffusion Evolution algorithm. The difference from the original Diffusion Evolution (Algorithm 1) is indicated in light blue.

---

**Algorithm 2** Latent Space Diffusion Evolution

---

**Require:** Population size $N$, parameter dimension $D$, latent space dimension $d$, fitness function $f$, density mapping function $g$, total evolution steps $T$, diffusion schedule $\boldsymbol{\alpha}$ and noise schedule $\boldsymbol{\sigma}$.

**Ensure:** $\alpha_0 \sim 1, \alpha_T \sim 0, \alpha_i > \alpha_{i+1}, 0 < \sigma_i < \sqrt{1 - \alpha_{i-1}}$

1:    $\boldsymbol{E}^{(d,D)} \leftarrow \mathcal{N}(0, 1/D)$          ▷ Initialize the random mapping

2:    $[\boldsymbol{x}_T^{(1)}, \boldsymbol{x}_T^{(2)}, ..., \boldsymbol{x}_T^{(N)}] \leftarrow \mathcal{N}(0, I^{N \times D})$          ▷ Initialize population

3:    **for** $t \in [T, T-1, ..., 2]$ **do**

4:       $\forall i \in [1, N] : Q_i \leftarrow g[f(\boldsymbol{x}_t^{(i)})]$          ▷ Fitness are cached to avoid repeated evaluations

5:       $\forall i \in [1, N] : \boldsymbol{z}_t^{(i)} \leftarrow \boldsymbol{E}\boldsymbol{x}_t^{(i)}$          ▷ Encode individual parameters into latent space

6:       **for** $i \in [1, 2, .., N]$ **do**

7:          $Z \leftarrow \sum_{j=1}^{N} Q_j \mathcal{N}(\boldsymbol{z}_t^{(i)}; \sqrt{\alpha_t}\boldsymbol{z}_t^{(j)}, 1 - \alpha_t)$

8:          $\hat{\boldsymbol{x}}_0 \leftarrow \frac{1}{Z} \sum_{j=1}^{N} Q_j \mathcal{N}(\boldsymbol{z}_t^{(i)}; \sqrt{\alpha_t}\boldsymbol{z}_t^{(j)}, 1 - \alpha_t)\boldsymbol{x}_t^{(j)}$

9:          $\boldsymbol{w} \leftarrow \mathcal{N}(0, I^D)$

10:         $\boldsymbol{x}_{t-1}^{(i)} \leftarrow \sqrt{\alpha_{t-1}}\hat{\boldsymbol{x}}_0 + \sqrt{1 - \alpha_{t-1} - \sigma_t^2} \dfrac{\boldsymbol{x}_t^{(i)} - \sqrt{\alpha_t}\hat{\boldsymbol{x}}_0}{\sqrt{1 - \alpha_t}} + \sigma_t \boldsymbol{w}$

11:       **end for**

12: **end for**

---

