# OpenReview forum: "Diffusion Models are Evolutionary Algorithms"
_ICLR.cc/2025/Conference — ICLR 2025 Poster_

### Official Review · Reviewer_tnPL · 2024-10-30

**Soundness:** 3
**Presentation:** 3
**Contribution:** 2
**Rating:** 8
**Confidence:** 4

**Summary:**

This paper explores the theoretical and practical parallels between diffusion models and evolutionary algorithms, proposing the "Diffusion Evolution" approach, which adapts diffusion models for evolutionary tasks. This new method, Latent Space Diffusion Evolution, enhances the ability to find diverse and optimal solutions in high-dimensional parameter spaces, particularly within reinforcement learning.

**Strengths:**

he core idea of treating diffusion models as evolutionary algorithms is innovative. It extends diffusion models to broader applications by framing them as tools for evolutionary tasks, potentially contributing new knowledge to both evolutionary biology and AI communities.

Technically: The paper provides thorough mathematical grounding for the equivalence between diffusion and evolution, specifically explaining diffusion as a probabilistic denoising process analogous to evolutionary mechanisms like mutation and selection.

Experimentals: Comprehensive experiments benchmark Diffusion Evolution against traditional algorithms (e.g., CMA-ES, PEPG) ascross various fitness landscapes, demonstrating its strength in maintaining diversity and achieving multiple optima.

Applications in High-Dimensional Problems: The adaptation of Diffusion Evolution to high-dimensional tasks via latent space diffusion showcases its practical viability in reinforcement learning contexts, with empirical results suggesting its potential in complex environments.

**Weaknesses:**

Complexity of Explanation: The theoretical connections between diffusion and evolution, while compelling, are highly complex, and the paper occasionally sacrifices clarity for depth. This might limit accessibility to a broader audience.

Although promising, the methodology may have limitations in scenarios requiring open-ended evolution, a challenge acknowledged briefly. More thorough discussion could help set realistic expectations for potential users.

While the Diffusion Evolution algorithm demonstrates superiority in finding diverse solutions, certain comparisons (e.g., high-fitness solutions) against traditional evolutionary algorithms lack statistical significance or detailed analysis of computational cost versus benefit, particularly in high-dimensional settings.

**Questions:**

The work potentially contributes new insights and methods for both diffusion model and evolutionary algorithm communities, even if the practical impact in specific real-world applications remains to be validated. Can the authors address this?

Can statistical tests be added?

---

> ### Author Response · Authors · 2024-11-21
>
> We thank you for your time and effort to review our manuscript and for your thoughtful comments on clarity, future directions, and high-dimensional statistics.
>
> We have revised the theoretical section to enhance its accessibility for a broader audience. We will update the paper soon.
>
> In the discussion section, we further explored the potential and limitations of our method. Specifically, regarding the challenge of open-ended evolution, several straightforward approaches can be directly adapted from the diffusion model domain. For example, the work by Yilun Xu et al., 2023 (https://arxiv.org/abs/2306.14878) introduces additional Gaussian noise to diffusion models, allowing their time steps to move backward. Their experiments demonstrate that this “restart” trick can improve quality. Based on the equivalence we established, we anticipate that such a method could facilitate open-ended evolution while also enhancing the quality of evolutionary results.
>
> For high-dimensional statistics, we extended our benchmarks to higher-dimensional spaces and provided more detailed statistical analyses across all benchmark experiments. Results on diversity and fitness are presented in this table:[ link](https://postimg.cc/ctYHbJ6M). For statistical significance, a detailed table including the standard deviation of fitness and diversity is available here:[ link](https://postimg.cc/N9Rvh27M). These analyses demonstrate that our superior performance in fitness and diversity is statistically significant.
>
> Additionally, we compared our method to the quality-diversity search method, specifically the MAP-Elite method, using the QD-score—a commonly used metric for assessing the balance between quality and diversity (see this paper https://arxiv.org/abs/2303.12803). The results show that our method, particularly the latent diffusion evolution, performs well across all tested fitness functions. Moreover, our method demonstrates significant improvements in QD-scores at high dimensions. See the detailed results in this table:[ link](https://postimg.cc/3dfkTr0T).
>
> **Addressing the reviewer’s concern about validation in real-world applications:**
>
> We have extended our reinforcement learning experiments to encompass a broader range of tasks, demonstrating that our method effectively identifies solutions across these RL scenarios. Additionally, we have incorporated statistical tests to strengthen the validation of our results. Please refer to the tables above. We will update the RL tasks in the revised paper soon.

---

### Official Review · Reviewer_HsP7 · 2024-11-02

**Soundness:** 3
**Presentation:** 3
**Contribution:** 3
**Rating:** 6
**Confidence:** 4

**Summary:**

This manuscript introduces an innovative perspective by interpreting the diffusion model as an evolutionary algorithm, highlighting the mathematical similarities between the diffusion and evolutionary processes. It proposes a Diffusion Evolution Method that employs iterative denoising to heuristically optimize solutions within the parameter space, drawing an analogy between the generative process of the diffusion model and the selection and mutation mechanisms in biological evolution.

**Strengths:**

The idea of regrading the diffusion model as an evolutionary algorithm is interesting.

**Weaknesses:**

1. The mathematical demonstration process in this manuscript is insufficient. The authors conceptualize the evolution process as a transformation of the probability density function, deriving the diffusion evolution algorithm from the Bayesian formula. However, this derivation overlooks key complexities inherent in evolutionary dynamics, such as genetic drift and gene recombination. Additionally, the application of the Bayesian formula assumes conditional independence, which may not be held in the context of evolutionary processes due to the interactions and competition among individuals. Furthermore, the manuscript does not provide a clear explanation of how to effectively map a complex fitness function into a probability density function, which is essential for the practical implementation of the proposed model.
2. Since the author claims that the diffusion model functions as an evolutionary algorithm, the experimental section must reflect this comparison appropriately. Specifically, in scenarios involving multiple tasks and comparisons, aspects such as speed, performance metrics (including best, worst, median, and average results), and the stability of performance should be examined. Additionally, the model's excellence should be illustrated through the mean variance of the results. Currently, the scope of experiments conducted is insufficient and does not adequately support the claims made in the manuscript.
3. There are several grammatical errors and unclear expressions throughout the article. For example, the definitions of certain terms lack clarity, the derivation process for the formulas is inadequately explained, and the labeling of the charts is inaccurate.
4. The authors categorize diffusion models as evolutionary algorithms on the basis that both methods perform distribution transformation. However, this classification may be overly broad. For instance, semantic segmentation models also involve transforming distributions—from real images to pixel-level segmentation results. Should we, therefore, classify these segmentation models as evolutionary algorithms as well? This comparison seems inaccurate. A more robust argument would establish that the Markov process within a non-equilibrium thermodynamics framework (diffusion) function as an unconstrained parameter optimization technique (evolution). If this argument cannot be substantiated, I recommend revising the title to better reflect the content.
5. The manuscript presents two primary sets of experiments: multi-objective evolution and latent space diffusion evolution. However, the multi-objective evolution experiments are limited to a two-dimensional parameter space and a simplistic fitness function, which fails to demonstrate the proposed approach's efficacy in high-dimensional parameter spaces or with more complex fitness functions. Similarly, the latent space diffusion evolution experiments focus solely on the CartPole task, lacking validation across a broader range of reinforcement learning tasks. Furthermore, the experimental results do not include any statistical significance tests, making it challenging to determine whether the observed improvements in algorithm performance are statistically significant. This omission significantly undermines the credibility of the findings presented.

**Questions:**

In the experiments section, the authors select three different evolutionary strategies for performance comparison. However, the rationale for choosing these specific methods is not clearly articulated. I would appreciate more insight into this selection process, particularly since each of these methods was introduced over five years ago.

---

> ### Author Response · Authors · 2024-11-22
>
> Thank you for such a detailed and thoughtful response.
>
> **Evolutionary Dynamics and Mathematical Foundations**
>
> Regarding the first concern about mathematical formulation, it's crucial to understand that we are not developing a new algorithm. Rather, our algorithm can be considered as the solution of an equivalence equation: we propose that $A(x) = B(x)$, by identifying the inherent similarities between $A$ and $B$. Here, $A$ is the diffusion model, and $B$ is the evolutionary algorithm. We then demonstrate our solution as the proposed method and establish its practical utility. Within this mathematical framework, we face natural constraints on incorporating arbitrary biological phenomenology. This is analogous to solving a simple equation like $x + 2 = 4$, where one cannot arbitrarily impose additional constraints, such as requiring $x$ to be negative.
>
> This mathematical perspective also addresses the concerns about conditional independence in Bayesian formulations. We're not explicitly modeling evolutionary process behavior, but rather deriving solutions from our equivalence assumptions and conditional independence.
>
> Nonetheless, our method still inherently incorporates several biological features without explicit design or imposition, **including the points you proposed**. Beyond the examples mentioned in the paper, such as reproductive isolation and natural selection, the Diffusion Evolution method also naturally encompasses genetic drift and gene recombination:
>
> 1. **Genetic Drift:** As suggested by the “neutral theory of molecular evolution,” many mutation dimensions do not impact fitness. This is precisely captured by learning or sampling the distribution—these neutral mutations broaden the distribution, facilitating genetic drift.
> 2. **Gene Recombination:** This feature is more explicitly included in Diffusion Evolution. In Equation 9, we estimate the optimal parameters as a weighted average of their neighbors. While biological recombination typically involves only two individuals per generation, our approach can be interpreted as a coarse-grained recombination over longer timescales. In the meantime, algorithms like CMA-ES are even doing gene recombination more implicitly and are still considered as evolutionary algorithms.
>
> We will add further discussions on these aspects in the revised paper and appreciate your feedback, which highlighted areas requiring clarification.
>
> **Choice of the density mapping function**
>
> Regarding the choice of the density function, it remains task-dependent. While evolutionary algorithms operate on nondifferentiable fitness functions, ensuring smooth probability functions and reasonable gradient values is advantageous, particularly for avoiding excessively small or large gradients. To address this, we are performing a grid search on benchmark functions. We modified the original function to a new form with a control temperature parameter. Higher temperatures smooth the density function, while lower temperatures focus on high-fitness parameters.
>
> We first calculate the fitness scale of the functions, defined as the maximum fitness difference around the optimal points in the fitness function. The temperature is then set to $T\times \text{scale}$, where $T$ is a parameter used to control the smoothness of the $g$ function. This rescaling enables the comparison of different fitness functions.
>
> Our observations reveal two general trends:
>
> 1. Higher temperatures lead the algorithm to sample more low-fitness parameters, resulting in a lower average fitness (see [[figure](https://postimg.cc/3dt8P3bH)]).
> 2. Increasing the temperature enhances result diversity, as reflected in both entropy (see [[figure](https://postimg.cc/V5244w4K)]) and quality-diversity scores—a metric quantifying the balance between quality and diversity (see definition at https://arxiv.org/abs/2303.12803). Our results are illustrated in [[figure](https://postimg.cc/Z0n61Q0X)].
>
> However, this effect plateaus quickly at approximately $T=1$. Based on these findings, we recommend first estimating the fitness scale and using $T=1$ for initial experiments. These guidelines will be incorporated into the revised paper shortly.

---

> ### Author Response · Authors · 2024-11-22
>
> **Statistical Analysis**
>
> Regarding point 2 in the weaknesses section, we appreciate your suggestion to include additional statistical analyses. While our primary objective was to establish the theoretical equivalence between diffusion and evolution processes—rather than emphasizing performance comparisons—our method demonstrates competitive performance even without specific optimization-focused design elements.
>
> In response to your recommendation, we have enhanced our evaluation in two ways.
>
> 1. First, we incorporated more detailed statistical analyses into the benchmark experiments. The low standard deviation of both fitness and diversity indicate that our superior performance compared to other methods is statistically significant and demonstrates stability. Please refer to the updated table 1 ([link](https://postimg.cc/ctYHbJ6M)) and the detailed table ([link](https://postimg.cc/N9Rvh27M)).
> 2. Second, we expanded our testing to include higher-dimensional fitness functions and comparisons with quality-diversity search approaches, which represent the cutting-edge method with impressive performance in evolutionary algorithms. Please also refer to the tables above. The results from these extended experiments further validate our method's capability to effectively balance exploration and exploitation, particularly when assessed using the quality-diversity score. See the comparison table at[ link](https://postimg.cc/3dfkTr0T).
>
> **Writing and grammar**
>
> For point 3 in the weakness section, we thank the reviewer for your time and effort. We carefully reviewed the paper and addressed grammar issues and other unclear points.
>
> **Distribution Transformations**
>
> Regarding point 4 in the weakness section, we agree that the concept of distribution is general and applies to various domains beyond evolution, such as prediction frameworks like the Gödel machine and general AI models like AIXI. However, this universality does not diminish the value of the distribution concept; rather, it provides numerous insights. Through the formulation of distribution transformation, we naturally connect fitness to probability density and obtain diversity. From the perspective of distribution, diversity implies distributions that are not narrow, such as a Dirac-delta function, or implies distributions with multiple peaks. Simultaneously, this concept ties directly to genetic drift, which you previously mentioned. Genetic drift is deeply related to neutral genes. In the framework of distribution, this corresponds to large regions in the parameter space with similar probability density.
>
> From a goal-oriented perspective, we aim to develop an evolutionary algorithm capable of finding multiple solutions, allowing for genetic drift, and modeling complex evolutionary processes. In such cases, viewing evolution as a distribution transformation proves to be the most effective approach.
>
> It is also worth noting that our connection between diffusion and evolution is not solely based on distribution transformations. Diffusion models inherently involve iterative refinement, sampling diverse solutions, and incorporating intrinsic random mutations. A fair comparison would reveal that approaches like semantic segmentation lack these characteristics.

---

> ### Author Response · Authors · 2024-11-22
>
> **Extending Experiments**
>
> Regarding point 5 in the weakness section, we appreciate your feedback on the experiments. Initially, we chose simple experiments because our primary focus was on establishing the theoretical connection between diffusion and evolution. However, this does not imply that our method lacks competitiveness. In the revised version, we extended our multi-objective evolution experiments to high-dimensional spaces, shown in the updated table 1 ([link](https://postimg.cc/ctYHbJ6M)). Additionally, we compared our approach to MAP-Elite, a popular quality-diversity search algorithm with impressive performance specifically designed to balance quality and diversity. These experiments demonstrate that our method, particularly the latent method, outperforms traditional evolutionary algorithms (EAs) and quality-diversity search approaches across various tasks, including high-dimensional ones. The comparison with quality-diversity search further highlights that our method surpasses MAP-Elite in effectively balancing quality and diversity. See the attached table for details ([link](https://postimg.cc/3dfkTr0T)). Such strong performance does not come from manually designed or imposed features like sampling, diversity, or genetic drift (which are nonetheless naturally encompassed as a consequence of the equivalence we discovered), which highlights the significant potential of this equivalence when applied to various tasks.
>
> Furthermore, we extended our reinforcement learning experiments to a broader range of tasks, including Acrobot, Cart Pole, Mountain Car, Continuous Mountain Car, and Pendulums. To broaden the scope of our testing, we introduced a scaling factor $s$. Specifically, given a default fitness function $f(x)$ for a reinforcement learning task, we transformed it into $f_s(x) = f(sx)$. This approach allows us to test tasks with varying norms of parameters, hence testing different topology of parameter distribution spaces.
>
> The experimental results, summarized in this table ([link](https://postimg.cc/HV0kY659)), indicate that for all environments, Diffusion Evolution and Latent Diffusion Evolution consistently achieve the highest rewards across different scaling factors. Moreover, the Latent Diffusion model demonstrates greater stability with respect to the scaling factor, validating the effectiveness of incorporating latent methods. Based on the standard deviations, these performance improvements are statistically significant. These results suggest that our method can serve as a general optimization approach.
>
> The statistical significance analysis on benchmark functions is discussed in previous sections.
>
> **Choice of Evolutionary Strategies**
>
> Lastly, addressing the issue raised in the question section. While these methods were introduced a long time ago, they were selected for comparison based on careful reasoning. Methods like CMA-ES and PEPG optimize parameters by modeling them as Gaussian distributions, and our comparison aims to demonstrate that the ability to fit complex distributions is critical for achieving both quality and diversity. The OpenES method was chosen because it also estimates the moving direction based on neighbors. The key differences are: (1) how the direction is estimated, and (2) in our method, each individual makes its own estimation, whereas OpenES relies on a single shared estimation. This comparison is intended to illustrate how a solution derived from the equivalence between diffusion and evolution can outperform methods that were meticulously designed.
>
> We have attached the updated tables and figures and will upload the revised paper shortly.

---

> > ### Comment · Reviewer_HsP7 · 2024-11-24
> >
> > Thank you to the authors for their thoughtful responses, which have provided me with a clearer understanding of the original article. Considering the methodological innovations presented in this paper and the authors' commitment to including more robust experimental results in the revised manuscript, I have decided to increase my score.

---

### Official Review · Reviewer_DCF9 · 2024-11-03

**Soundness:** 2
**Presentation:** 2
**Contribution:** 2
**Rating:** 5
**Confidence:** 4

**Summary:**

This paper proposes a new evolutionary algorithm inspired by diffusion models. It views the evolution processes as the denoising process of diffusion models, and designs an evolutionary algorithm named Diffusion Evolution, which is based on the denoising framework of the famous diffusion model DDIM. During evolution, it takes each individual in the population as a solution in the denoising process, and updates it with similar updating rules in DDIM. To facilitate better performance, it further follows previous works to optimize in the latent space. Experiments on several benchmark functions and a simple cart-pole controlling problem show that compared with classic baseline methods include CMA-ES, OpenES and PEPG, the proposed Diffusion Evolution can obtain more diverse solutions with good fitness.

**Strengths:**

The paper is well-written and easy to follow. The proposed method Diffusion Evolution is well-motivated, with clear explanation and illustration.

**Weaknesses:**

As there has been a variety of evolutionary algorithms with various inspirations, the most important issue when proposing a new method should be clarifying its strength compared with previous methods, from the perspectives of theoretical analysis and extensive experiments. However, the proposed Diffusion Evolution in this paper is lack of theoretical analysis. Meanwhile, the experiments are much too simple. For example, only five synthetic functions and a simple cart-pole controlling problem are included. As the diversity of solutions seems to be the strength of the proposed Diffusion Evolution, except for methods like CMA-ES, which are not specified for diversity, comparison with other kinds of evolutionary algorithms like Quality-Diversity (QD) are necessary.

**Questions:**

As mentioned in the appendix, different Alpha and noise schedule settings are tested, what about the detailed experimental results? Is there any analysis about the results of different settings, which could be useful for the users?

---

> ### Author Response · Authors · 2024-11-21
>
> Thank you for your thoughtful critique. We presented this paper in a theoretical framework,  and one of our primary contributions lies in its theoretical framework  specifically in demonstrating the mathematics of the equivalence between diffusion and evolution  processes. Having rigorously established this equivalence, we can directly apply existing theoretical analyses of diffusion models - including convergence properties - without the need for additional theoretical investigation. This fundamental equivalence enables us to import techniques from diffusion model research, particularly accelerated sampling methods. These methods demonstrate the same acceleration in our evolutionary framework without requiring modifications to other components. The demonstrated equivalence serves as the cornerstone of our theoretical analysis, making such adaptations possible
>
> For the benchmark experiments, we selected CMA-ES from a distributional perspective. Since CMA-ES represents a distribution using a Gaussian, it lacks the capability to fit and optimize over complex fitness landscapes, such as those with multiple optima. This allowed us to highlight the flexibility of our method in addressing such challenges. We also agree that incorporating quality-diversity search can enhance the benchmark comparison. To this end, we integrated MAP-Elite into our benchmark and extended the fitness function to higher dimensions. The results showed that our method outperformed MAP-Elite. While maintaining comparable diversity (quantified using entropy) with MAP-Elite, our method achieved significantly higher fitness, demonstrating its ability to balance exploration and exploitation through unbiased sampling.
>
> We thank the reviewer for their suggestions to improve our presentation and better highlight our method's advantages over other evolutionary algorithms. We have expanded our evaluation by testing our approach on additional reinforcement learning tasks, which demonstrated its capability to discover effective controllers while maintaining low computational overhead. We included more experiments and statistics and showed that our method can outperform traditional methods on both quality and diversity. Especially in QD-score (https://arxiv.org/abs/2303.12803), we outperform all other methods on high dimensional space, including MAP-Elite, a popular quality-diversity search. Please refer to the [QD-score table](https://postimg.cc/3dfkTr0T)
>
> Regarding the question on alpha scheduling, we appreciate your inquiry. Generally, we observe similar behavior to that in the diffusion model domain. For instance, cosine alpha scheduling significantly impacts convergence speed in diffusion models, and we observe the same effect in diffusion evolution when applying cosine alpha scheduling. Our experimental results, presented in [figure](https://postimg.cc/PPXscxLX), compare fitness values across different total evolution times under various alpha scheduling strategies. Compared to the default DDPM alpha scheduling and linear decreasing alpha scheduling, cosine alpha scheduling outperforms the other methods when the number of evolution steps is below 100. While the performance difference becomes less significant with a higher number of steps, the primary motivation for adopting cosine alpha scheduling is to reduce the total number of evolution steps required.
>
> We are finalizing the analysis and will upload the paper soon.

---

> ### Author Response · Authors · 2024-12-02
>
> Dear Reviewer,
>
> Thank you for your detailed feedback on our paper. We have conducted extensive additional experiments to address your concerns about our method's performance and theoretical foundations. Please check our revised PDF manuscript, changes highlighted with blue color.
>
> ## Comprehensive Statistical Analysis
> We have expanded our evaluation across various fitness functions and comparison methods. For performance and diversity metrics, we conducted two key experiments. In the fitness evaluation, our method demonstrated strong performance while maintaining diversity, even when tested in higher dimensions (4, 32, 256) as shown in Tables 1 and 3. Following your suggestion, we compared our approach with MAP-Elite using the QD-score metric. The results show our method outperforms MAP-Elite in high-dimensional spaces while remaining competitive in lower dimensions (Table 4).
>
> We've also broadened our reinforcement learning experiments to include Acrobot, Cart Pole, Mountain Car (both discrete and continuous), and Pendulum environments. Our method consistently outperforms CMA-ES across all tasks (Table 5) with robust success rates (Table 6). To test robustness, we introduced a scaling hyper-parameter that modifies the fitness landscape, and our method maintained consistent performance across these variations.
>
> ## Theoretical Connections
> Our new experiments provide stronger support for our theoretical framework. We observed interesting parallels between diffusion and evolution processes. The alpha scheduling behavior shows remarkable similarities in both contexts - the cosine scheduling, originally proposed for diffusion models, demonstrates similar benefits in our evolutionary context when total evolution time is limited (Figure 5). Additionally, the latent method's contribution to performance mirrors patterns seen in diffusion models, particularly in addressing high-dimensional challenges (Tables 1 and 6).
>
> Through grid search experiments (Figure 6), we've found that fitness function design follows principles similar to gradient-based training: optimal performance occurs when the fitness function is neither too sensitive nor too insensitive to parameter changes. We propose using an energy-based method (Equation 19) with temperature T as a tunable parameter to achieve this balance.
>
> -------
>
> We believe these additional results address your concerns about both empirical performance and theoretical foundations. We would greatly appreciate your thoughts on these new findings.

---

> ### Author Response · Authors · 2024-12-03
> **Final Additional Theoretical Discussion**
>
> Thank you for your thoughtful review. We would like to further elaborate on the **theoretical aspects** you raised.
>
> Regarding diversity, we have identified intriguing connections between Latent Diffusion Evolution and MAP-Elite. Both methods effectively avoid global competition, resulting in diverse solutions. In MAP-Elite, fitness comparisons occur only within a given classification. Similarly, in Diffusion Evolution, the Gaussian term in Equation 9 ensures that evolution and fitness comparisons are localized. This means that a population with highly distinct parameters does not affect the evolution of others too much, even if it possesses very high fitness. Our Latent Diffusion Evolution extends this concept by making such comparisons more flexible. While we currently use random mapping for the latent mapping function, other meaningful mappings can also be employed. This approach is akin to MAP-Elite but achieves diversity through a smoother mechanism. Instead of entirely preventing comparisons between distinct populations, our method reduces the magnitude of competition among them. This smoothness property may provide advantages in certain applications and could explain why our method outperformed MAP-Elite in QD-scores at high dimensions (Table 4 in the revised version) - each individual has more information about the fitness landscape.
>
> On robustness, we conducted extensive tests by modifying the fitness function in reinforcement learning tasks. Our experiments scaled the fitness landscape across four orders of magnitude ($10^4$), and the results demonstrated remarkable stability across these parameter variations (see Tables 5 and 6).
>
> Regarding convergence, we acknowledge its significance. However, the core contribution of our paper lies in establishing the equivalence between diffusion models and evolutionary algorithms. To preserve this fundamental equivalence, we intentionally refrained from introducing additional optimization tricks. A key implication is that **our method inherits the convergence properties of diffusion models to desired distributions**.
>
> We believe your participation in the discussion phase would help ensure the rating accurately reflects our paper’s contributions, particularly in light of these theoretical clarifications.

---

### Official Review · Reviewer_jGHU · 2024-11-04

**Soundness:** 3
**Presentation:** 3
**Contribution:** 3
**Rating:** 8
**Confidence:** 3

**Summary:**

The authors try to connect Diffusion Models and Evolutionary computation by arguing that both processes do iterative refinements through an update rule plus some perturbation. In evolutionary algorithms the update comes in the form of natural selection, in diffusion is the denoising phase. The perturbation corresponds to mutation for evolution, and the diffusion phase for diffusion models.

From the above connection an evolutionary algorithm that performs in its iteration a diffusion process is proposed.  Instead of aiming to recover some distribution of the data, the goal is to turn the random initial population points towards a distribution centered around the optimized function optimum.

**Strengths:**

By connecting methods from different fields the work proposed an interesting new evolutionary algorithm that can be further improved using variations and techniques found in diffusion models.

The text is very clear and the math derivation from diffusion to an EA algorithm is easy to follow. Figure 1 really helps to visualize how the population of the diffusion EA spreads to the higher fitness regions.

**Weaknesses:**

Maybe I missed it but there is no discussion on how to choose the mapping to a density function g() or which g() was used for the experiments. It seems is important for the search to work properly to have a mapping that makes it clear which points should be paid more attention by assigning to them bigger weights. Maybe there is some connection to common selection strategies in evolutionary algorithms that could be discussed.

**Questions:**

In the text is not mention what g could be or some options. It could be like the utility function used in Natural Evolutionary Strategies? because it seems to assign more weight/importance to high ranking solutions.

I think it would be very interesting in a  future work to explore more the latent diffusion version of the algorithm, and techniques to understand which parameters could be ignored or which parameters should be perturbed together.

---

> ### Author Response · Authors · 2024-11-21
>
> We appreciate your acknowledgment of our novelty and for pointing out several important items for us to address. In general, the choice of the g function is task-dependent. While evolutionary algorithms can operate with non-differentiable fitness functions, making the probability functions smooth and ensuring reasonable gradient (or some kind of pseudo gradients) values can be beneficial—specifically by avoiding excessively small or large gradients. To further address this question, we conducted a grid search on the benchmark functions. We modified the original g function to a new form with a control temperature parameter. Higher temperatures make the density function smoother, whereas lower temperatures make it more focused on high-fitness parameters.
>
> We first calculate the fitness scale of the functions, defined as the maximum fitness difference around the optimal points in the fitness function. Then, we set the temperature to be $\text{scale}\times T$, where $T$ is the parameter used to control the smoothness of the $g$ function. This rescaling allows for the comparison of different fitness functions.
>
> We observe two general trends: (1) Higher temperatures lead the algorithm to sample more low-fitness parameters, resulting in a lower average fitness (see [figure](https://postimg.cc/3dt8P3bH)). (2) Increasing the temperature enhances result diversity, as reflected in both entropy (see [figure](https://postimg.cc/V5244w4K)) and quality-diversity scores – a metric that quantifies the balance between quality and diversity (see definition at https://arxiv.org/abs/2303.12803). Our results are shown in [figure](https://postimg.cc/Z0n61Q0X). However, this effect plateaus quickly at around $T=1$. Therefore, we recommend first estimating the fitness scale and then using $T=1$ for initial experiments. We will update these guidance to the revised paper shortly.
>
> We also appreciate your last question and find it intriguing. As discussed in the paper, biological systems naturally employ multiple mechanisms to reduce dimensionality and assign importance to certain parameters, a concept known as sloppy model. Beyond this, geometric location and various non-genomic features also play a role. These directions will be examined in forthcoming work.

---

> ### Author Response · Authors · 2024-12-02
>
> Dear Reviewer,
>
> Thank you again for your thoughtful comments on our work. We have revised our PDF manuscript with all changes highlighted in blue for easier reference.
>
> We've conducted analysis on the $g$ function evaluation that you inquired about, as shown in Figure 6 in revised version. We assessed its impact across three key metrics: fitness, diversity, and QD score. Our findings suggest a principle similar to gradient-based neural network training - the fitness sensitivity should be carefully balanced. If it's too high, diversity suffers; if it's too low, fitness improvement becomes challenging.
>
> Your question about the latent space prompted us to explore an interesting extreme case. We investigated what happens when we initially introduce latent variables that are completely independent of the parameter space. Specifically, we expanded a 2D parameter space by appending two additional dimensions:
> $$
> (x_1,x_2)\to(x_1,x_2,{\color{blue}x_3,x_4})
> $$
> This approach mimics geometric distances found in biological systems. The results on the two-peaks fitness function (where (-1,-1) and (1,1) are optimal points) revealed an intriguing phenomenon of spatial isolation in parameters: we observed distinct clusters forming in the additional 2D latent space, where each cluster contains similar gene parameters, see [link](https://postimg.cc/v1xY8Kby) here. This finding not only provides insights into how geometry influences evolution but also suggests a potential clustering method.
>
> We believe these new results offer additional perspective on your questions about the role and behavior of latent spaces in our method. We would appreciate your thoughts on these findings.

---

> > ### Comment · Reviewer_jGHU · 2024-12-03
> >
> > Thanks to the authors for their responses.
> > I appreciate the addition of the probability mapping function while it may be dependent on the task, giving an example makes easier for other people to replicate your work.
> > The comment related to the latent space, I appreciate the authors took the time to investigate it now. Hope this develops further in a separate work.
> >
> > The manuscript improved by these additions and the ones from other reviewers' comments so I decided to raise my score.

---

### Author Response · Authors · 2024-11-28
**General Reply**

We thank the reviewers for their insightful feedback and suggestions for improving our manuscript. All comments have been addressed in our revised manuscript, with changes highlighted in blue. Below we summarize our responses to the main concerns raised.

**General Purpose and Contributions:** While we have demonstrated and compared our performance with other methods on various benchmarks and reinforcement learning tasks, our primary focus remains on the equivalence we identified between diffusion and evolution. We emphasize this point because such equivalence offers significant potential both for computer science and biology. On the methodological side, it establishes a new bridge between the fields of diffusion and evolution, enabling mutual contributions and performance improvements, as demonstrated by our accelerated sampling and latent diffusion evolution, enhanced population diversity, and ability to maintain multiple solutions simultaneously. On the theoretical side, this equivalence opens up novel perspectives for studying both diffusion and evolution.

Although this work primarily represents a discovery rather than a new method, we have not compromised on performance. Notably, our approach discovers diverse solutions across various tasks and even outperforms the MAP-Elite method in balancing quality and diversity, despite not being specifically designed for these features.

**Benchmark and Statistics:** As suggested by reviewers, we extended our benchmark experiments to higher dimensions and incorporated additional statistical measurements. Please refer to Tables 1 and 3 for detailed results and statistics. The extended experiments further validate our original claims: (1) our diffusion evolution method discovers diverse solutions; (2) while the original Diffusion Evolution may struggle with high-dimensional problems, the Latent Diffusion Evolution significantly improves performance and outperforms other methods in both diversity and fitness. We also expanded the reinforcement learning tasks to demonstrate the effectiveness of our method across additional RL tasks. Please refer to Tables 5 and 6 for detailed statistics.

**Comparison with Quality-Diversity Search:** We further compared our method with quality-diversity search, specifically the MAP-Elite algorithm, which is designed to balance quality and diversity. While MAP-Elite performs better on 2D experiments in terms of QD-score, our Latent Diffusion Evolution significantly outperforms MAP-Elite on high-dimensional problems, even without explicitly incorporating such features. Please refer to Table 4.

**Probability Mapping Functions:** We provided a general guide for choosing the probability mapping function (i.e., the $g$ function). We modified the original $g$ function by introducing a $T$ parameter to control its sharpness and performed a grid search on the $T$ parameter. In general, a good probability mapping function should have appropriate gradients, neither too steep nor too shallow, even though our method does not require differentiable probability functions. A practical guideline is to first estimate the standard deviation of fitness values, then rescale it to have unit standard deviation. The temperature $T$ can typically be set around 1, depending on whether diversity or fitness is prioritized. Please refer to Figure 6 for details.

**Writing and Introductory/Discussion Sections:** We carefully reviewed and improved grammar and writing issues. The Introduction was revised to enhance clarity and flow. In the Discussion section, we elaborated on the limitations of our method and proposed potential solutions.

> Please note that there may be discrepancies between the final revised version and the tables included in the previous reply. This is due to further improvements in the implementation code and the addition of random seeds for reproducibility.

---

### Meta-Review · Area_Chair_phVc · 2024-12-20

**Metareview:**

This paper bridges the fields of machine learning and evolutionary computation by revealing a mathematical equivalence between diffusion models and evolutionary algorithms. Diffusion models, initially designed for generative tasks, are reinterpreted as performing evolutionary algorithms through iterative denoising. The denoising process in diffusion models is analogous to natural selection in evolutionary algorithms, while the perturbation step corresponds to mutation or diffusion. Based on this insight, the authors introduce a novel evolutionary algorithm called Diffusion Evolution.

Diffusion Evolution uses the iterative denoising process from diffusion models to refine solutions in a parameter space. Instead of recovering a data distribution, the algorithm shifts an initial random population toward an optimized solution distribution. By leveraging latent space diffusion and accelerated sampling techniques, the authors further propose Latent Space Diffusion Evolution, which addresses high-dimensional optimization problems with reduced computational overhead. This method enables the efficient discovery of diverse and high-quality solutions.

Experimental results on benchmark optimization functions and a cart-pole control task demonstrate that Diffusion Evolution outperforms traditional evolutionary algorithms like CMA-ES, OpenES, and PEPG. It achieves improved solution diversity and fitness, showcasing its effectiveness in complex optimization scenarios.

The reviewers have generally found the contribution sound, novel, and worthy of publication and I highly recommend this paper for being accepted.

**Additional Comments On Reviewer Discussion:**

There have been constructive discussions among the authors and reviewers, which have led to addressing the key concerns raised by some of the reviewers.

---

### Decision · Program_Chairs · 2025-01-22

Accept (Poster)